# Conjugation dynamics depend on both the plasmid acquisition cost and the fitness cost

Hannah Prensky[1], Angela Gomez-Simmonds[2], Anne-Catrin Uhlemann[2] & Allison J Lopatkin[1,3,4,*] 

## Abstract

Plasmid conjugation is a major mechanism responsible for the spread of antibiotic resistance. Plasmid fitness costs are known to impact long-term growth dynamics of microbial populations by providing plasmid-carrying cells a relative (dis)advantage compared to plasmid-free counterparts. Separately, plasmid acquisition introduces an immediate, but transient, metabolic perturbation. However, the impact of these short-term effects on subsequent growth dynamics has not previously been established. Here, we observed that *de novo* transconjugants grew significantly slower and/or with overall prolonged lag times, compared to lineages that had been replicating for several generations, indicating the presence of a plasmid acquisition cost. These effects were general to diverse incompatibility groups, well-characterized and clinically captured plasmids, Gram-negative recipient strains and species, and experimental conditions. Modeling revealed that both fitness and acquisition costs modulate overall conjugation dynamics, validated with previously published data. These results suggest that the hours immediately following conjugation may play a critical role in both short- and long-term plasmid prevalence. This time frame is particularly relevant to microbiomes with high plasmid/strain diversity considered to be hot spots for conjugation.

**Keywords** antibiotic resistance; conjugation; fitness cost; horizontal gene transfer; mathematical modeling

**Subject Categories** Metabolism; Microbiology, Virology & Host Pathogen Interaction

**Mol Syst Biol. (2021) 17: e9913**

## Introduction

Horizontal gene transfer (HGT) by conjugation, which refers to the transfer of DNA from a donor to a recipient through direct cell-to-cell contact (Frost & Koraimann, 2010) (Fig 1A), is the predominant way bacteria mobilize and exchange antibiotic resistance genes (Maiden, 1998; Barlow, 2009). Conjugation can occur via the transfer of chromosomally integrated conjugative elements or autonomously replicating plasmids. Plasmids, which often encode one or multiple antibiotic resistance genes (Holmes *et al*, 2016), are primarily responsible for the global dissemination of resistance since approximately half of all plasmids are conjugative (Smillie *et al*, 2010) and can have broad host ranges (Klumper *et al*, 2015). Moreover, conjugation via plasmid transfer is postulated to be prevalent in complex microbial communities (e.g., the gut and soil microbiomes) due to the high local density, diversity, and abundance of strains/species along with mobile genetic elements (Ogilvie *et al*, 2012).

Conjugation dynamics depend on the formation and proliferation of new HGT progeny (termed transconjugants) and are fundamentally governed by two kinetic processes: the rate of gene transfer (termed the conjugation efficiency) and the relative growth rate of transconjugants (termed the growth dynamics). Both the conjugation efficiency and growth dynamics may depend on a host of extrinsic and intrinsic factors. For example, the cell's physiological state can drastically alter the conjugation efficiency of certain plasmids by orders of magnitude (Lopatkin *et al*, 2016b). Likewise, growth dynamics are highly dependent on both the functional benefit and metabolic burden of a given plasmid, which can yield net positive or negative effects on relative growth rates. Measuring both processes has enabled accurate predictions of plasmid persistence in simple, *in vitro* HGT communities, often consisting of one or few plasmids and strains, thereby improving our overall understanding of plasmid dynamics (Lopatkin *et al*, 2017). However, native bacterial communities consist of hundreds of diverse plasmids and species that interact, grow, and compete over multiple time scales (Jorgensen *et al*, 2014). Predicting plasmid fate and overall conjugation dynamics in these more complex settings remains a challenge (Dunn *et al*, 2019; Lopatkin & Collins, 2020).

It is widely known that, in the absence of antibiotic selection, plasmids can exert a fitness cost on their hosts. Fitness costs can vary widely and are often attributed to the metabolic burden of the plasmid. A plasmid's fitness cost is typically quantified using an established lineage (e.g., isolated transconjugant clone), in direct competition assays with, or as a relative growth rate compared to, the plasmid-free counterpart (Ponciano *et al*, 2007). Previous studies have shown that plasmid fitness costs can alter population

1   Department of Biology, Barnard College, New York, NY, USA
2   Division of Infectious Diseases, Department of Medicine, Columbia University Irving Medical Center, New York, NY, USA
3   Department of Ecology, Evolution, and Environmental Biology, Columbia University, New York, NY, USA
4   Data Science Institute, Columbia University, New York, NY, USA
    *Corresponding author. Tel: +1 212 853 2564; E-mail: alopatkin@barnard.edu

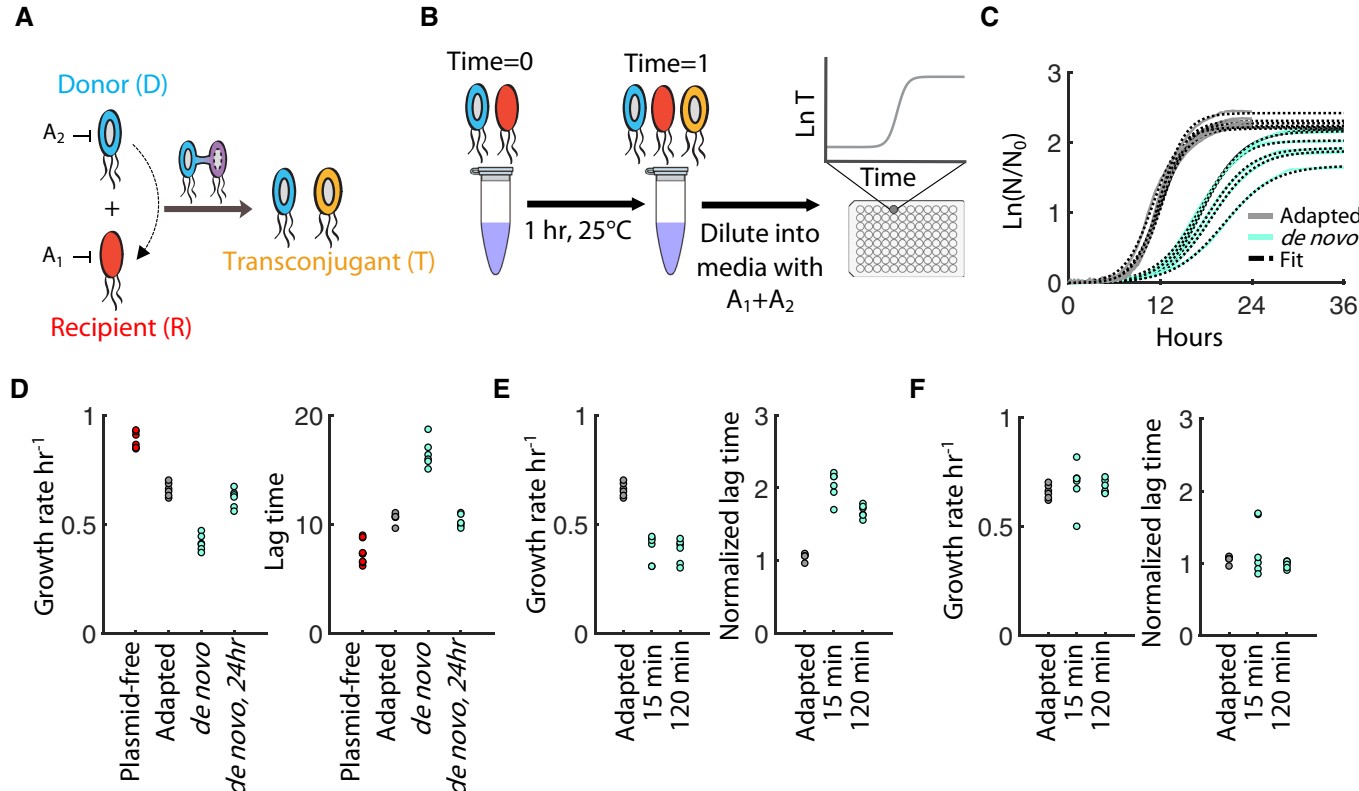

**Figure 1.** Transconjugants exhibit an acquisition cost following conjugation.

A Schematic of conjugation whereby plasmid DNA from a donor (D, blue) is transferred to a recipient (R, red), generating a transconjugant (T, yellow). R and D are each resistant to an antibiotic ($A_1$ and $A_2$, respectively), but sensitive to the other. The transconjugant (T) is uniquely resistant to both antibiotics.

B Conjugation protocol involves mixing D and R, followed by one-hour incubation at 25°C. Cells are then diluted into media containing $A_1$ and $A_2$ and growth is captured over time in a 96-well plate.

C $OD_{600}$ for de novo T (aqua) and adapted T (gray), (RP4 transconjugants) initiated with the same number of cells per well. Each curve is a biological replicate. Black dashed lines are best-fits.

D Growth rate (left) and lag time (right) for the plasmid-free recipient R, adapted T, de novo T after 1 h of conjugation, and de novo T diluted and re-grown after 24 h. De novo T growth rates (aqua) are statistically less than adapted T (gray) ($P = 1.12e-08$, Appendix Table S2) and plasmid-free cells (red) ($P = 7.27e-09$, Appendix Table S2). De novo T lag times are statistically greater than adapted T ($P = 5.71e-08$ Appendix Table S2) and plasmid-free cells (red) ($3.77e-09$, Appendix Table S2). Data represent biological replicates. All statistics were done using a one-way ANOVA with Bonferroni correction.

E Left: De novo T growth rates (aqua) are statistically less than adapted T growth rates (gray) ($P = 7.34e-05$ and $4.95e-05$ for 15 and 120 min, respectively). Right: Lag times were multiplied by true $T_0$ and divided by the mean adapted T lag (Appendix Fig S1D is non-normalized). De novo T (aqua) normalized lag times are statistically less than adapted T (gray) ($P = 1.30e-09$ and $1.31e-09$ for 15 and 120 min, respectively). Data represent biological replicates. All statistics were done using a one-way ANOVA with Bonferroni correction.

F After 24 h, each condition (E) was diluted and re-grown. Left: growth rates are statistically identical ($P = 1.00$ for both 15 and 120 min). Right: All lag times are statistically identical ($P = 0.69$ and $P = 0.48$ for 15 and 120 min, respectively). Lag times normalized as described in (E). All data represent biological replicates. All statistics were done using a one-way ANOVA with Bonferroni correction.

Source data are available online for this figure.

structure and dynamics; for example, costly plasmids can be outcompeted, leading to elimination, or compensated for by mutations that ameliorate the metabolic burden, prolonging plasmid persistence over time (Dahlberg & Chao, 2003; Dionisio, 2005; Harrison et al, 2015; Loftie-Eaton et al, 2017).

Separate from fitness costs, acquiring a plasmid via conjugation requires immediate physiological adaptation (e.g., altered gene regulation and/or resource allocation (San Millan et al, 2018)) and therefore also impacts cellular metabolism. For example, it has been shown that plasmid-encoded stress response genes are transiently expressed in the host for 20–40 min following plasmid acquisition (Althorpe et al, 1999; Baharoglu et al, 2010); the SOS response accounts for a considerable component of bacterial maintenance metabolism (Kempes et al, 2017). Likewise, recent work demonstrated that the overshoot of plasmid-encoded gene expression only occurs in recently generated transconjugants (Fernandez-Lopez et al, 2014). The extent of metabolic disruption following conjugation suggests that plasmid acquisition may also impact growth dynamics. However, compared to the fitness cost, considerably less work has focused on these immediate effects, which we refer to as the plasmid acquisition cost. Indeed, these impacts are not captured in traditional fitness cost measurements since they occur soon after

conjugation and quickly stabilize (Fernandez-Lopez & de la Cruz, 2014); they may also manifest in diverse growth effects, such as altered growth rates and/or lag time preceding exponential growth, which renders quantification challenging. Overall, the generality and magnitude of plasmid acquisition costs remain widely unknown. Given the diversity and abundance of plasmids in natural environments, quantifying plasmid acquisition costs may shed insights into plasmid fates in mixed/competing populations.

Here, we discovered that newly generated transconjugants exhibited transient but overall reduced growth rates and/or prolonged lag times, indicating the presence of an increased burden immediately following conjugation—a plasmid acquisition cost. This plasmid acquisition cost occurs independently of long-term fitness effects, potentially corresponding to the initial energetic burden imposed by establishing a newly acquired plasmid, as well as the host cells' ability to efficiently fulfill that burden in a given environment. Moreover, incorporating acquisition effects into a mathematical model of conjugation improved temporal predictions of long-term conjugation dynamics across a range of plasmids. These results demonstrate the prevalence and importance of short-term metabolic effects in conjugation dynamics and have implications in understanding and predicting dominant plasmids in the environment.

## Results

### Acquisition of RP4 impacts growth dynamics

To investigate how plasmid acquisition might affect transconjugant growth, we sought to compare the growth dynamics of *de novo* transconjugants (which have not undergone physiological adaptation) to those previously established (and therefore fully adapted). *De novo* transconjugants (T) can be readily generated using our previously established protocol for estimating conjugation efficiencies (Lopatkin *et al*, 2016a): donors (D) and recipients (R) carrying unique resistance genes are mixed under conditions that minimize growth. Since T is uniquely resistant to both antibiotics, it can then be directly selected from the population, and its growth tracked over time in a microplate reader (Fig 1B). This procedure is ideal for our purposes since it minimizes growth and adaptation during the conjugation window, ensuring that subsequent dynamic characterization of *de novo* T captures emergent phenotypic changes. In contrast, adapted T can be isolated by streaking conjugation mixtures onto dual antibiotic agar plates; individual clones can then be grown and stored for subsequent testing (Dahlberg & Chao, 2003; Rozwandowicz *et al*, 2019). These established transconjugants exhibit stably reproducible growth rates and are often used to quantify fitness costs and/or determine the timescale of compensatory mutations (Harrison *et al*, 2015; Hall *et al*, 2020).

Using this approach, we first focused on the well-established, large conjugal plasmid RP4 (~60 kb). We chose RP4 since initial characterization revealed it as imposing a fitness cost on the cell, which allows us to distinguish between immediate acquisition costs and compensatory mutations thereafter (Appendix Fig S1A). Briefly, D and R were established using *Escherichia coli* MG1655 strains (Appendix Table S1A); R expresses spectinomycin (Spec) resistance, and D, which carries the RP4 plasmid, encodes kanamycin (Kan) resistance. To measure the growth of *de novo* T, D and R were

mixed for one hour at 25°C, diluted 1,000× into Spec-Kan liquid media, and tracked via $OD_{600}$ in a microplate reader. In parallel, adapted T clones were incubated under identical conditions (e.g., one hour, 25°C) to control for any physiological effects of the protocol itself, and subsequently diluted into Spec-Kan liquid media at a comparable initial cell density, as verified with colony forming units (CFU) (Appendix Fig S1B).

Strikingly, *de novo* T appeared to grow overall slower than adapted T (Fig 1C). Indeed, curve-fitting using the established Baranyi equation (Baranyi & Roberts, 1994) revealed that *de novo* T's growth rate was significantly lower, and the lag time significantly higher, than that of adapted T (Fig 1D, Appendix Fig S1C, $P = 1.12e\text{-}08$ and $P = 5.71e\text{-}09$, respectively). These results were independent of the conjugation duration: mixtures incubated for both 15 and 120 min exhibited similar trends (Fig 1E and Appendix Fig S1C). However, when these populations were diluted and re-grown after 24 h, both the growth rate and lag time were fully restored (Fig 1F). Importantly, in all cases, the recovered growth rates remained lower than that of the plasmid-free strain, indicating the plasmid retained its fitness cost (Fig 1D). Finally, these results were independent of the method used to quantify growth parameters since three additional quantification methods resulted in qualitatively consistent trends (Appendix Fig S2, Appendix Table S2).

Although these results initially suggest that RP4 acquisition is costly, we identified several protocol-related factors that could potentially account for these observations. First, the protocol requires *de novo* T to adjust to a new growth environment, possibly altering growth dynamics independently from conjugation-specific effects. However, adapted T was subjected to identical experimental conditions, which accounts for any effect of environmental adaptation on growth independent of conjugation. Second, competition with residual R/D cells could alter *de novo* T growth. To test this, we initiated growth of adapted T with R/D diluted 1,000× in the background media; this approximates the parental densities present during the conjugation experiment. Doing so did not qualitatively affect the trajectory of adapted T, nor the growth discrepancy between adapted and *de novo* T (Appendix Fig S1D). Moreover, there was no appreciable background conjugation of R/D at this density (Appendix Fig S1E), indicating that neither population survived long enough to conjugate during this time window. Overall, we conclude that RP4 is indeed costly to acquire, and that plasmid acquisition can modulate both the growth rate and lag time.

### Introducing a quantitative metric for the plasmid acquisition cost

That RP4 acquisition induced transient changes in both growth rate and lag time is intuitive: The reduced growth rate is consistent with fitness cost interpretations as a metabolic burden associated with replication/protein expression. Moreover, a cell's immediate response to a metabolic perturbation is known to manifest in altered lag dynamics, such as following a nutrient shift (Madar *et al*, 2013). Therefore, to facilitate quantification, we sought to define a rigorous and accurate plasmid acquisition metric that would capture all effects of plasmid acquisition on growth dynamics. To this end, a minimal model of cell growth suggested that the time required to reach a predetermined threshold density is an inclusive proxy for

changes in both growth rate and lag time (Fig 2A and Appendix Fig S3); this is consistent with a recent study that used an analogous "time to threshold" method to compare conjugation efficiencies *in vitro* (Bethke *et al*, 2020).

Briefly, let T(t) describe the growth of transconjugants over time, and μ be the maximum specific growth rate. Consistent with previous literature (Métris *et al*, 2006), extrapolating the exponential growth region to the horizontal axis allows us to define the geometric lag time (λ), which corresponds to the observable onset of exponential growth (Fig 2A). During the exponential phase, $T(t)$ can thus be described by the line: $\ln(T(t)) = \ln(T_0) + \mu(t-\lambda)$, where $T_0$ is the true initial population density. Under these conditions, the time ($t^*$) it takes the population to reach a specific detection level ($T^*$) is

inversely correlated with $T_0$. In other words, we can predict an unknown initial cell density ($T_{pred}$) from its observed time to threshold ($t^*$) value, assuming $T^*$, μ, and λ are constant (Fig 2B). Conversely, a discrepancy between $T_{pred}$ and true $T_0$, which CFU can determine, specifically indicates that μ and/or λ have changed. For our purposes, consider a standard curve relating $t^*$ and $T_0$ using adapted T, and a $T_{pred}$ generated from a *de novo* T population as described above. In that case, any discrepancy between $T_{pred}$ and true $T_0$ (e.g., $T_{pred}/T_0 < 1$) can be attributed to growth-specific consequences of acquiring a plasmid. Moreover, the use of $T_0$ as a metric of acquisition cost, rather than $t^*$, allows us to simultaneously compare conjugation rates across diverse experimental conditions and plasmids. Additionally, generating an *a priori* standard

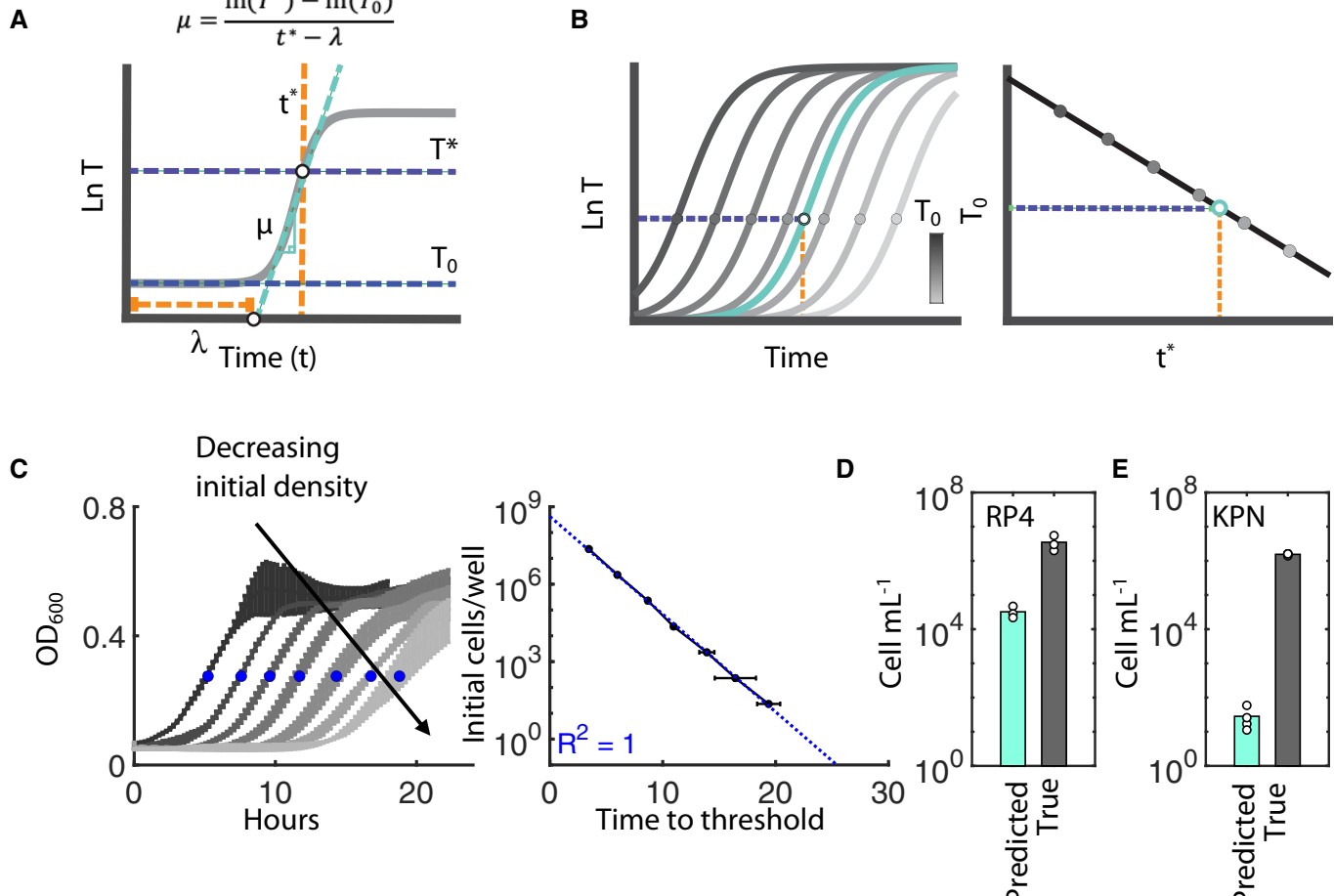

**Figure 2. Acquisition cost quantification for RP4.**

A   The time ($t^*$, orange) at which a specified cell density threshold ($T^*$, top purple) is reached uniquely depends on the initial cell density ($T_0$, bottom purple), and the growth rate (μ, aqua) and lag times (λ, orange). Assuming background subtraction, the line can be represented by the equation that is shown.

B   Representative standard curve generation is shown. *Left*: To generate the standard curve, adapted T are diluted in 10-fold increments and growth is measured over time (dark to light gray is $T_0$). Circles indicate the $t^*$ (purple line) corresponding $T^*$ (orange line). Aqua line represents the out-growth of T following a conjugation experiment (i.e., *de novo* T). *Right*: The initial cell density is plotting against $t^*$; black line indicates the standard curve.

C   *Left*: RP4 adapted T growth initiated with decreasing true $T_0$ (dark to light gray); blue markers show the time to reach $OD_{600}$ of 0.275 ($t^*$). *Right*: Standard curve is shown in blue. Error bars are standard deviation of three biological replicates.

D   True and predicted CFU of RP4 with the recipient *E. coli* strain MG1655. Scatter points represent three biological replicates, and bar height is the average.

E   True and predicted CFU of RP4 with the recipient *K. pneumoniae* (KPN) strain AL2425. Scatter points represent four biological replicates, and bar height is the average.

Source data are available online for this figure.

curve spanning a broad range of initial densities also avoids the variability inherent in manually diluting adapted $T$ to a specific target number, as we had done initially (Fig 2B). As such, predicted compared to true CFU represents a more robust and trustworthy quantification method.

We verified the utility of this metric by confirming it could capture the observed discrepancies in RP4 growth. Specifically, we built a standard curve using adapted T (Fig 2C) and predicted the initial *de novo* T population densities ($T_{pred}$) based on times to threshold quantitated from observed growth curves. Having previously verified the true $T_0$ with CFU counts, we found that $T_{pred}$ was significantly less than true $T_0$ ($P = 0.0143$, right-tailed $t$-test), indicating an acquisition cost for RP4 (Fig 2D), as expected. We note that a standard curve generated with R and D diluted $1,000\times$ in the background media to approximate the parental densities present during the conjugation experiment does not significantly change the observed discrepancy between true $T_0$ and $T_{pred}$ (Appendix Figs S4A and B), consistent with earlier.

### Generality of the plasmid acquisition cost

To investigate the generality of the acquisition cost, we first determined whether it was unique to our particular experimental conditions. Specifically, we compared true $T_0$ and $T_{pred}$ estimates for RP4 using different experimental parameters (e.g., dilution factor, conjugation time window, recipient strain). Results revealed that the RP4 acquisition cost was independent of the conjugation time window as shown previously (Appendix Fig S4C i), recipient strain (Appendix Fig S4C ii), and the dilution factor (Appendix Fig S4C iii-iv); indeed, dilution factors of $150\times$, $500\times$, $1,000\times$, and $5,000\times$ predicted a separate but parallel standard curve (Appendix Fig S4D), indicating a systematic difference between the two estimates. Moreover, RP4 was costly to acquire for the clinically isolated recipient strain *Klebsiella pneumoniae*(KPN), indicating species-level generality (Fig 2E). The drastic difference in RP4 acquisition costs between *E. coli* and KPN recipients suggest that cost is not solely a function of particular plasmids; strain/species-level attributes are likely key as well.

Next, we reasoned that the long replication time of the large RP4 plasmid (~60 kb), coupled with the significant amount of energy required to synthesize conjugation machinery upon acquisition, likely imposed an immense energetic burden that led to transient growth inhibition, and thus, a high acquisition cost (San Millan & MacLean, 2017). Conversely, we hypothesized that mobilizable plasmids, which are transferred by conjugation *in trans* but do not themselves encode conjugation machinery, would most likely result in a minimal acquisition cost. The absence of genes encoding for conjugation machinery reduces both the plasmid size and the burden of the machinery production. To test this, we used the $F_{HR}$ helper plasmid system described previously (Dimitriu *et al*, 2014), which is not self-transmissible but can mobilize any co-residing plasmid that encodes the recognition sequence *oriT*. We chose the mobilizable plasmid pR which carries chloramphenicol resistance. As with RP4, this plasmid exhibits an overall fitness cost (Appendix Fig S5). Consistent with our hypothesis, post-conjugation growth curves overlapped with those used to determine the standard curve (Fig 3A). Indeed, experiments revealed a statistically identical match between true $T_0$ and $T_{pred}$ (Fig 3B, $P = 0.34$ and

0.86 for two- and one-tailed $t$-tests, respectively). Thus, we conclude that pR does not induce a significant acquisition cost. More generally, these results confirm that observed acquisition costs, as determined by the difference in transconjugant estimates ($T_0$ and $T_{pred}$), are not artifacts of the experimental method and can be detected using this quantification metric.

Given that RP4 and pR-specific differences likely arise due to differences in energetic demand, we hypothesized that altering growth efficiency (e.g., the amount of substrate consumed that is converted to biomass) (Chudoba *et al*, 1992) would modulate acquisition costs. Intuitively, inefficiently growing cells generate excess available energy (Russell & Cook, 1995; Russell, 2007) that may be readily applied to plasmid-related metabolic demands, potentially resulting in a lower acquisition cost. In contrast, efficiently growing cells devote the bulk of available energy to biomass production (Low & Chase, 1999), and thus, reallocating that energy to plasmid demands may increase acquisition costs. To test this hypothesis, we focused on modulating growth conditions. It is well-established that excess glucose yields highly inefficient *E. coli* growth (Liu, 1998; Basan *et al*, 2015), but that efficiency is restored with exogenous amino acid supplementation (Akashi & Gojobori, 2002; Waschina *et al*, 2016). Adapting a recent approach that leveraged this trade-off (Lopatkin *et al*, 2019), we quantified plasmid acquisition costs under three casamino acid (CAA) concentrations (0, 0.01, and 0.1% w/v) and excess glucose (0.4% w/v). Since higher CAA increases both efficiency and growth rate, we included a fourth combination (0.04%/0.1% w/v glucose/CAA) where growth rate is comparable to 0.4%/0.01% w/v, but results in higher efficiency due to the lower glucose level (Fig 3C). In accordance with our intuition, increased CAA resulted in significant decreases in acquisition costs (Fig 3D; $P < 0.05$, one-tailed $t$-test, see Appendix Table S3B for exact $P$-values). Moreover, this trend was not an effect of increased growth: acquisition cost in 0.04%/0.1% glucose/CAA was significantly less than in 0.4%/0.01% glucose/CAA conditions ($P = 0.04$, one-tailed $t$-test). Together, these results suggest that environmental conditions significantly modulate acquisition cost through changes in growth efficiency.

We next looked at acquisition costs across a broad panel of plasmids to further investigate its generality. Specifically, we first quantified acquisition costs for six well-characterized conjugal plasmids (R1, incF; R1drd, incF; R6K, incX; R6Kdrd, incX; pRK100, incF; RIP113, incN; Appendix Table S1B) covering four additional incompatibility groups and a range of fitness costs (Appendix Fig S5). Importantly, two pairs of these plasmids represent derepressed mutants and their native repressed counterparts (R1 and R6K). Although both plasmid types express conjugation machinery immediately following their transfer, repressed plasmids tightly regulate machinery expression shortly thereafter, minimizing their fitness costs (Lundquist & Levin, 1986) (Appendix Fig S5). Interestingly, we observed no qualitative differences in R1 or R6K acquisition costs, regardless of conjugation repression (Fig 3E): Both R6K/R6K-drd, and neither R1/R1drd, were costly to acquire. Since all variants express machinery immediately upon entry, these results suggest that repression of conjugative machinery occurs on a timescale longer than that of acquisition cost for these four plasmids. Rather, we noted acquisition cost differences across incompatibility groups. Specifically, all four incF plasmids (R1, R1drd, and pRK100), along with pR from earlier, imposed no significant acquisition cost,

whereas R6K and R6Kdrd (incX) and RIP113 (incN) induced a strong acquisition cost. Incompatibility groups are differentiated by their plasmid replication/partitioning mechanisms as well as specific copy number (Kittell & Helinski, 1993); for example, incF plasmids typically exist at low copy numbers ($\leq\sim$2) (Burger *et al*, 1981), whereas incX plasmids can be present at 10–15 copies per cell (Rakowski & Filutowicz, 2013). Thus, these results suggest that acquisition costs may arise as a consequence of establishing plasmid-specific replication and maintenance mechanisms.

Finally, we quantified the acquisition cost for four clinical self-transmissible plasmids previously obtained from pathogenic isolates encoding extended-spectrum β-lactamase (ESBL) enzymes (Lopatkin *et al*, 2017). These enzymes confer resistance to a wide range of β-lactam drugs. As with the well-characterized plasmids, these encompass a range of fitness costs, including one that proved beneficial to the host (Appendix Fig S5). Despite this diversity, $T_{pred}$ was consistently and significantly lower than true $T_0$ for all ESBL plasmids (Fig 3F, $P < 0.05$, one-tailed *t*-test, see Appendix Table S3A for

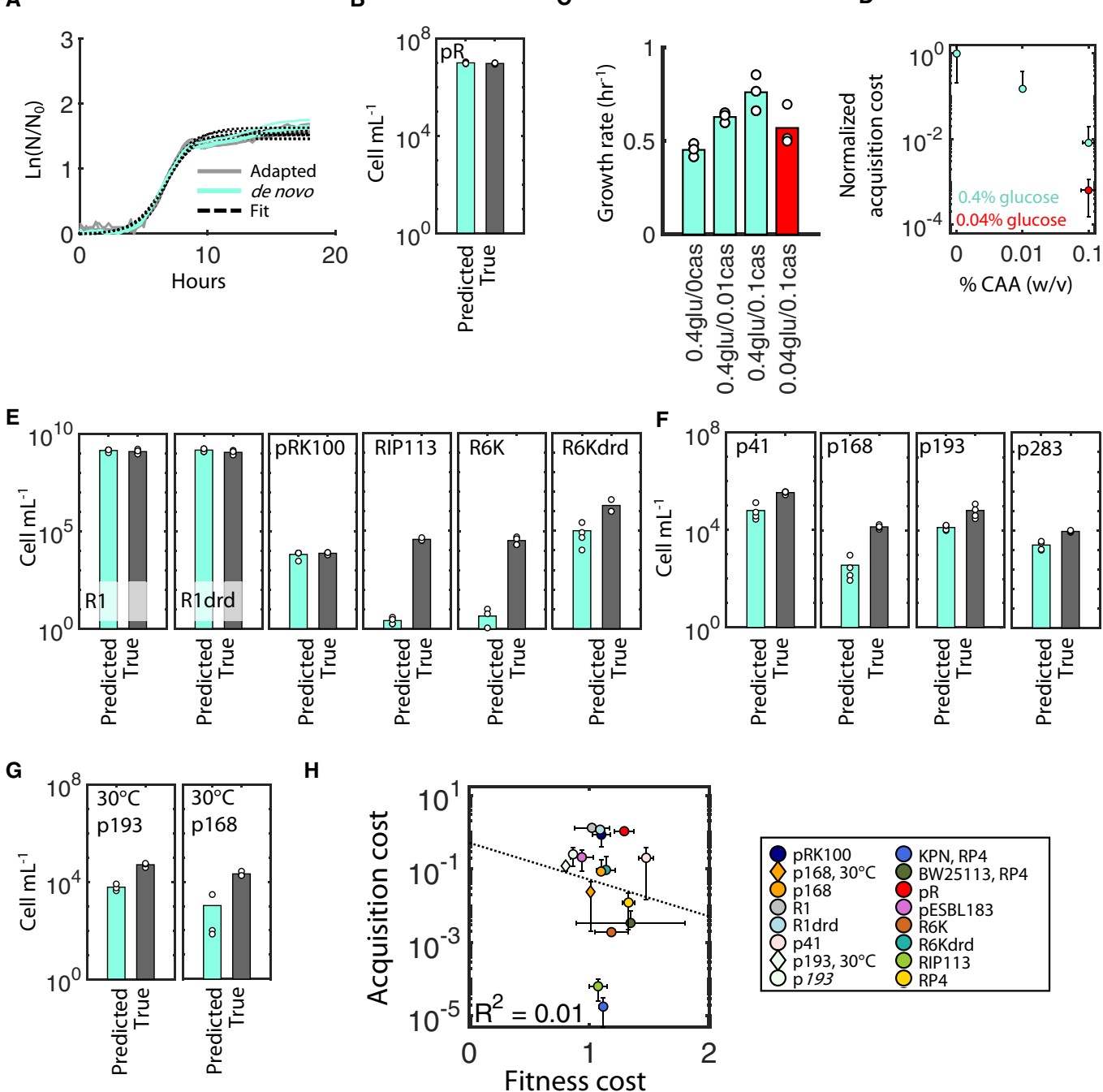

Figure 3.

**Figure 3.   Generality of acquisition cost.**

A   $OD_{600}$ for *de novo* (aqua) and adapted transconjugants (gray) for the plasmid pR are shown over time. Black lines are best-fits. Individual curves are biological replicates.

B   True and predicted CFU for the plasmid pR are statistically identical ($P = 0.34$ and $0.86$ for one and two-tailed *t*-tests, respectively). Scatter points represent biological replicates, and bar height is the average.

C   Growth rates are shown for adapted T carrying RP4 under variable glucose (glu) and casamino acid (caa) concentrations. Values represent % w/v. Scatter points represent biological replicates.

D   Acquisition costs were quantified for the same glucose and casamino acid concentrations from (C). Scatter points represent the average, and error bars represent standard deviation, of three biological replicates. Aqua and red indicate glucose at 0.4% and 0.04% w/v, respectively. Y-axis is acquisition cost ($T_{pred}/T_0$) normalized to the cost in the absence of casamino acids.

E   $T_{pred}$ compared to $T_0$ for six well-characterized plasmids. Representatives are shown of two biological replicates (see Appendix Fig S10 for day-to-day variability). R1, R1drd, and pRK100 do not show a significant acquisition cost ($P = 0.93$, $0.79$, and $0.28$, respectively), whereas RIP113, R6K, and R6Kdrd do ($P = 6.79e\text{-}05$, $7.57e\text{-}05$, and $0.037$, respectively, one-tailed *t*-test, Appendix Table S3A).

F   $T_{pred}$ was significantly less than $T_0$ for clinical plasmids p41, p168, p193, and p283 at 37°C ($P = 7.10e\text{-}05$, $2.10e\text{-}05$, $0.021$, $1.90e\text{-}05$, respectively, $n = 4$, $4$, $3$, $2$, respectively, one-tailed *t*-test, Appendix Table S3A). In all cases except p283, bars represent averages and scatter points individual measurements from at least three biological replicates; p283 has two biological replicates.

G   $T_{pred}$ for two clinical plasmids, p193 and p168, at 30°C was significantly less than $T_0$ ($P = 2.80e\text{-}04$, $2.72e\text{-}04$, respectively, $n = 3$, $4$, respectively, one-tailed *t*-test, Appendix Table S3A).

H   All acquisition costs scattered against the fitness cost measured under the identical condition for each plasmid. Acquisition costs are measured as the ratio between $T_{pred}/T_0$. Black line is the linear regression line of best fit, and $R^2 = 0.01$ (shown in the bottom left). Error bars represent standard deviation; the type of replicates used for these error bars is listed in Appendix Table S3A.

Source data are available online for this figure.

exact *P*-values). This finding held true at lower out-growth temperatures and for different donor/recipient pairs as well (Fig 3G). Moreover, we observed a significant acquisition cost even when the overall fitness cost was beneficial, as was the case with p193. Overall, we conclude that different conjugal plasmids can incur unique and varying acquisition costs, the magnitudes of which are likely dependent on a host of environmental and host-specific factors.

We summarized all acquisition costs as the ratio between $T_{pred}$ and true $T_0$ and compared them to fitness costs to determine whether there was any significant relationship between the two measurements. Combined, these cover 12 plasmids and five incompatibility groups (Appendix Table S1A). Doing so revealed a statistically insignificant relationship between the two variables (Fig 3H), leading us to conclude that although the two costs may have similar underlying constraints, they are ultimately imposed, at least in part, by independent factors.

**Mathematical model of conjugation that accounts for *de novo* transconjugants**

Thus far, we have shown that transconjugants may exhibit a reduced growth rate and overall prolonged lag time immediately following conjugation. These altered growth dynamics returned to expected levels within 24 h; however, steady-state growth rates indicated that the plasmid retained its fitness cost, suggesting that compensatory mutations had likely not occurred. While these short-term experiments enabled us to rigorously quantify plasmid-dependent growth effects, they do not necessarily provide insights into longer-term population dynamics. Indeed, over extended periods, *de novo* transconjugants are continually generated, leading to potential competitive effects and non-homogeneous adaptation within a mixed population. To capture and elucidate this additional complexity and better describe how acquisition cost modulates overall population structure, we modified a previously published model of conjugation (Lopatkin *et al*, 2017) to further investigate the impact of transient physiological plasmid adaptation on both short- and long-term dynamics (Appendix Equations S1-S2; Appendix Fig S6A).

In the simplest case, consider a population S that either does ($S^1$) or does not ($S^0$) have the conjugal plasmid (Appendix Fig S6A). $S^0$ gains the plasmid from $S^1$ at a rate constant, the conjugation efficiency ($\eta$), thereby becoming $S^1$. Likewise, $S^1$ can lose the plasmid at a rate constant associated with plasmid segregation error ($\kappa$), thus transitioning back to $S^0$. Critically, the growth dynamics in this original model are governed primarily by the relative fitness cost (e.g., $\mu_1 = \mu$, and $\mu_0 = \alpha\mu$), where $\mu_1$ and $\mu_0$ are the growth rates of $S^1$ and $S^0$, respectively, and $\alpha$ is a scalar that represents the plasmid fitness impact: $\alpha > 1$ implies the plasmid is costly to the host, whereas $\alpha < 1$ denotes benefit.

We previously used this model to investigate the conditions favoring plasmid persistence in mixed populations as a function of only the conjugation efficiency and fitness cost (Lopatkin *et al*, 2017). However, plasmid acquisition cost can readily be incorporated into this framework. In particular, we assume that $S^1$ (e.g., all plasmid-carrying cells) consists of the sum of both *de novo* ($S^D$), as well as fully adapted ($S^A$), transconjugants (Fig 4A, equations 1–3):

$$\frac{dS^0}{dt} = \alpha\mu S^0\left(1 - S^0 - S^A - S^D\right) - \eta S^0 S^A + \kappa S^A - DS^0 \quad (1)$$

$$\frac{dS^D}{dt} = \rho\mu S^D\left(1 - S^0 - S^A - S^D\right) + \eta S^0 S^A - \beta S^D - DS^D \quad (2)$$

$$\frac{dS^A}{dt} = \mu S^A\left(1 - S^0 - S^A - S^D\right) + \beta S^D + \kappa S^A - DS^A \quad (3)$$

Here, conjugation between $S^0$ and $S^A$ results in the formation of $S^D$, which subsequently transitions into the fully adapted transconjugant population $S^A$ at a plasmid-specific transition rate $\beta$. As in the original model, all populations grow logistically with a common carrying capacity and are diluted at a rate D. Moreover, $S^D$ grows at a rate relative to $S^A$ given by $\mu_D = \rho\mu$, where $\rho$ is a scalar between 0 and 1. Thus, as with our experimental data, the combined effect of $\rho$ and $\beta$ accounts for the plasmid acquisition cost. Finally, based on our data that acquisition costs were not qualitatively different

between derepressed plasmids and their natively repressed counterparts, we assume that both conjugation and plasmid loss from $S^D$ is negligible compared to $S^A$; however, we note that this assumption does not qualitatively change the modeling results nor impact our main conclusions (Appendix Fig S6C).

Intuitively, the presence of $S^D$ should not drastically alter the qualitative behavior of the expanded model compared to the original. Indeed, when both $S^0$ and $S^A$ are initially present, $S^D$ accumulates over short time scales (e.g., 24 h), but is limited due to carrying capacity constraints (Appendix Fig S6B). Rather, $S^D$ introduces a temporal delay in overall transconjugant dynamics. Specifically, depending on the values of $\rho$ and $\beta$, $S^D$ growth decreases the overall growth rate of all plasmid-carrying cells (e.g., $S^1 = S^D + S^A$), thereby delaying peak growth and steady-state density. This interpretation of plasmid acquisition cost is fully consistent with our experimental methods, since it allows for changes in both growth rate and lag times. Moreover, our measurements did not distinguish between $S^D$ and $S^A$.

To investigate whether this model could account for our observed experimental results, we first sought to determine whether the expanded model could recapitulate the RP4 dynamics. In particular, we assume that both dilution and plasmid segregation are negligible, consistent with this batch growth setup (Appendix Table S4). We then fit the remaining unknown parameters ($\rho$ and $\beta$) to the entire transconjugant population ($S^1$) using a simulation initiated with 100% *de novo* transconjugants ($S^D$) (Fig 4 B). Doing so resulted in quantitatively accurate predictions of both observed growth rates and lag times (Fig 4C).

We next sought to quantitatively examine the impact of $\rho$ and $\beta$ on the observed growth rate of $S^1$ to better understand their individual effect. Since $S^D$ transitions to $S^A$, the observed growth rate of $S^1$ ($\mu_{obs}$) can be defined as the weighted average of both transconjugant populations (Appendix equation S6), and therefore changes over time as transconjugants are newly generated and adapted. Simulations revealed that, for a fixed $\beta$, a faster *de novo* transconjugant growth rate allows the observed growth rate to remain higher

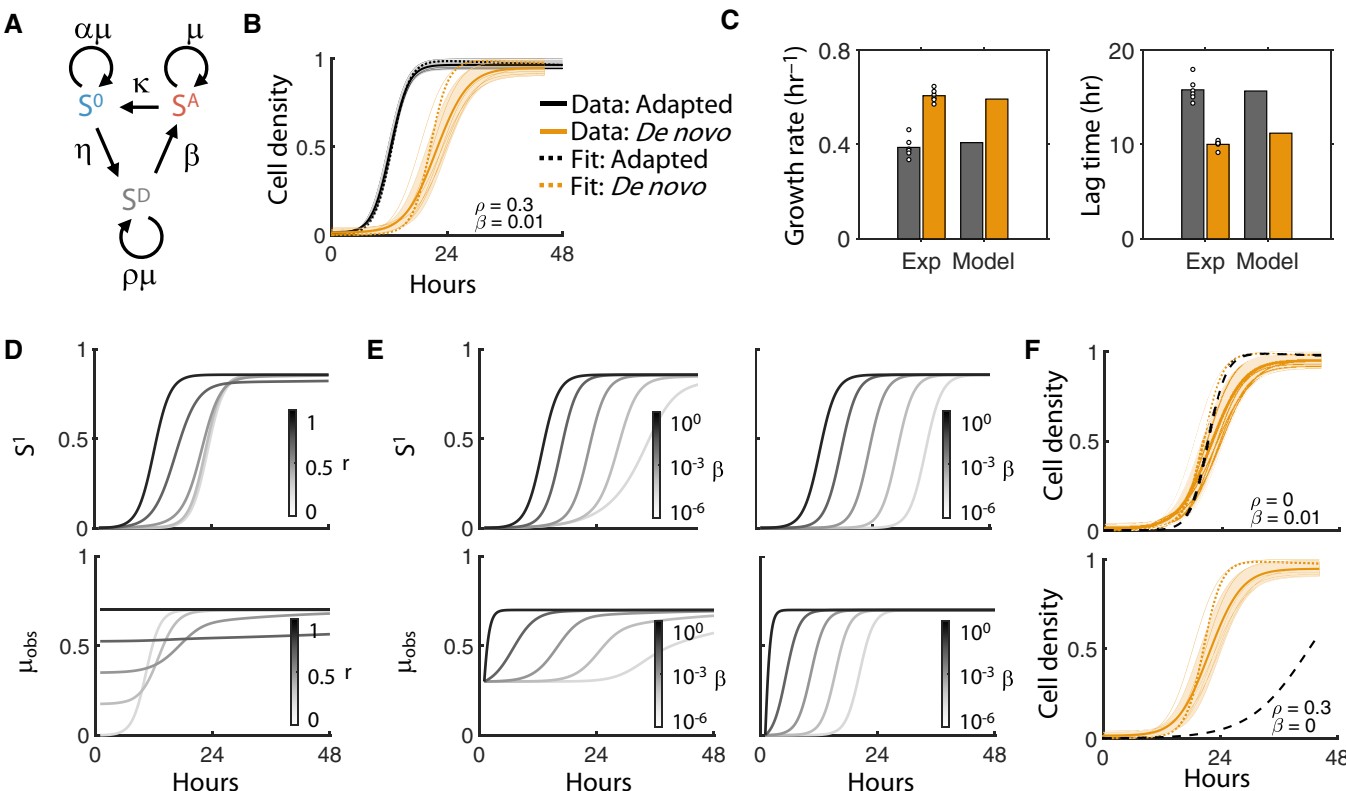

**Figure 4. Conjugation model incorporating acquisition cost.**

A   Network diagram of conjugation model. The plasmid-free population ($S^0$) acquires the plasmid from the plasmid-adapted population ($S^A$), turning into a transient *de novo* transconjugant ($S^D$) at a rate $\eta$ (the conjugation efficiency). Finally, $S^A$ can revert to $S^0$ according to the plasmid segregation error rate $\kappa$. The *de novo* population, in turn, transitions into the adapted population at the rate $\beta$. Growth rates for $S^0$ and $S^D$ are scaled relative to the plasmid-adapted population ($\mu$) based on the scalars $\alpha$ and $\rho$, respectively. Not included in diagram: dilution of all populations out of the system at rate D.

B   RP4 data were fit to the model to calculate $\beta$ and $\rho$. Dotted line shows model fit. Data are from Fig 1C.

C   Model predicts accurate growth rates and lag times based on fitted parameters.

D   Parameter sensitivity to $\rho$. Bottom: Average observed growth rate is defined as $(S^D\rho\mu + S^A\mu)/(S^D + S^A)$, and is measured for increasing $\beta$ from light to dark. Top: Corresponding population density of $S^1$ over time. $\beta$ is fixed to 0.01 based on RP4 fitting.

E   Parameter sensitivity to $\beta$. Bottom: Average observed growth rate measured for increasing $\beta$ from light to dark where $\rho = 0.3$ (left) or $\rho = 0$ (right). Top: Corresponding population density of $S^1$ over time.

F   RP4 data (Fig 4B) are re-fit with fixed $\rho$ (top) and fixed $\beta$ (bottom).

throughout the entire duration (Fig 4D *bottom*), and $S^1$ is modulated accordingly (e.g., growing faster with increasing ρ, Fig 4D *top*). Likewise, for a fixed ρ, $S^1$ reached the maximum growth rate (e.g., that set by $S^A$) more rapidly as the transition rate (β) increased; this corresponds to growth inhibition when transition is slow (Fig 4E *left*). Interestingly, though, even when *de novo* transconjugants cannot grow (ρ = 0), the observed growth rates largely followed the expected β trajectories, reaching the maximum level within ~24 h (Fig 4E *right*). These simulations suggested that the transition rate (β) rather than the scaled growth rate (ρ), may be primarily responsible for fitting accuracy. Indeed, fitting either β or ρ individually revealed that β was sufficient to predict the RP4 growth dynamics, while ρ was not (Fig 4F). Based on these observations, we considered β the primary driver of the acquisition cost for subsequent investigation into its interplay with fitness costs.

## Incorporating plasmid acquisition costs better predict temporal conjugation dynamics

As described above, it is well-established that fitness costs impact population structure and temporal dynamics in heterogeneous communities over time. To this end, we recently used the original model (Appendix Fig S6A, Appendix equations S1-S2) to accurately predict plasmid fate in mixed communities of plasmid-free and plasmid-carrying cells; we found that even costly plasmids could persist with a sufficiently rapid conjugation rate (Lopatkin *et al*, 2017). This model, however, was not always able to quantitatively capture temporal behaviors for the range of plasmids we tested. Given that plasmid acquisition and subsequent adaptation is a continuous process, we reasoned that our expanded model may provide further insight and accuracy toward predicting long-term conjugation dynamics. Indeed, simulations initiated with identical conditions as in our previous setup (e.g., initial $S^D = 0$) revealed that $S^D$ may still significantly contribute to the overall population structure when an acquisition cost is present (Fig 5A). Thus, we next sought to use our expanded model to determine the relative contribution of both fitness and acquisition costs: To what extent do both these processes modulate overall temporal population dynamics?

To address this question, we first examined temporal dynamics in the limiting case described above (Fig 4E), i.e., that *de novo* transconjugants accumulate and transition, but do not grow (ρ~0). We chose this scenario initially due to the challenge of experimentally measuring *de novo* growth rates in isolation (i.e., quantifying ρ). Therefore, the long-term fitness cost of carrying the plasmid remains (α) and the acquisition cost is approximated by the transition rate, β. Here, we note that this formulation preserves the observed reduction in overall transconjugant growth rates. Under these conditions, increasing the transition rate increases the adapted fraction (compared to *de novo* cells), with β = 1 approximating our original model (i.e., $S^D \approx 0$ and $S^A \approx S^1$).

Sensitivity analysis revealed that both the fitness cost (α) and acquisition cost (β) modulate the time until the steady state of $S^1$ was reached (Fig 5B). To quantify the contribution from each cost, we calculated the difference between the potential maximum growth rate of $S^1$ (μ, as set by $S^A$), and the observed steady-state growth rate of $S^1$ ($\mu_{obs}$, Appendix equation S6), which may be less than μ, since it consists of the combined populations $S^D$ and $S^A$. In the case where the observed and maximum growth rates are equivalent (μ = $\mu_{obs}$), β does not impact overall $S^1$ growth, and the effect of acquisition cost on the temporal dynamics is negligible compared to fitness cost. Conversely, any deviation (i.e., $\mu_{obs}$/μ < 1) indicates a β-specific effect. Indeed, a boundary defined by α and β delineates the parameter regions where observed growth deviates from the maximum growth (Fig 5C). Specifically, when the acquisition cost is sufficiently low, growth of $S^1$ is largely unaffected by $S^D$; therefore, fitness cost alone will likely capture the temporal dynamics. However, when the acquisition cost is high, $S^D$ does impact the growth of $S^1$, particularly for costly plasmids; in this case, acquisition costs should be incorporated to improve temporal predictions. This conclusion was maintained when ρ > 0 (Appendix Fig S6D).

Consistent with the original model, the boundary dictating these scenarios is dependent on the conjugation efficiency (η): when conjugation is sufficiently slow, neither acquisition nor fitness costs alter $\mu_{obs}$/μ. Intuitively, in this case, since there is minimal accumulation of $S^D$, *de novo* transconjugants do not appreciably change the overall growth rate. Surprisingly, however, rapid conjugation is not necessarily beneficial to the plasmid-carrying population. If generation of $S^D$ via conjugation is significantly faster than the transition from $S^D$ to $S^A$ (i.e., η>>β), the observed growth rate decreases; this counterintuitive finding suggests that faster conjugation may not favor plasmid retention unless paired with correspondingly rapid adaptation.

These results give rise to two general scenarios of plasmid persistence. First, if the observed and maximum growth rates are approximately equal ($\mu_{obs} \approx \mu$), the acquisition cost has a minimal effect and α is sufficient to predict overall dynamics (Fig 5C, red region); this roughly corresponds to α < 1, but can include regions where α > 1 if the conjugation efficiency is sufficiently fast. Moreover, as predicted by our original model, a fast conjugation efficiency can overcome a plasmid burden (α > 1), thus accounting for the expanding red region as a function of η. Second, as $\mu_{obs}$ diverges from μ (Fig 5C, blue region), increasing fitness costs amplify the effect of β; here, acquisition cost impacts growth rate and thus population dynamics.

To test these predictions, we re-examined previously generated data whereby transconjugants were co-cultured with the corresponding plasmid-free population over the course of 14–21 days. In each case, we independently fit β with short-term growth data (Appendix Fig S7) and simulated long-term dynamics (Fig 5D). We found that our experimental data were indeed fully consistent with the expanded model predictions. In particular, although p193 was predicted to have a high acquisition cost, incorporating it does not significantly affect predictive accuracy as the plasmid is beneficial. Conversely, for plasmids that did have a fitness cost (e.g., RP4, p41, and p168), the magnitude of β correlated with whether it was needed for predictive accuracy: plasmids with lower β diverged more strongly from the original model and were better predicted by the updated one that accounted for its effects. Here, we note that changing the value of ρ such that it remained low did not appreciably alter simulation results. Collectively, these results validate our model and demonstrate the potential impact acquisition costs may have on long-term population dynamics. For example, p41 and p168 exhibited similar fitness costs; however, the high acquisition cost of the latter would likely favor the former in a mixed population.

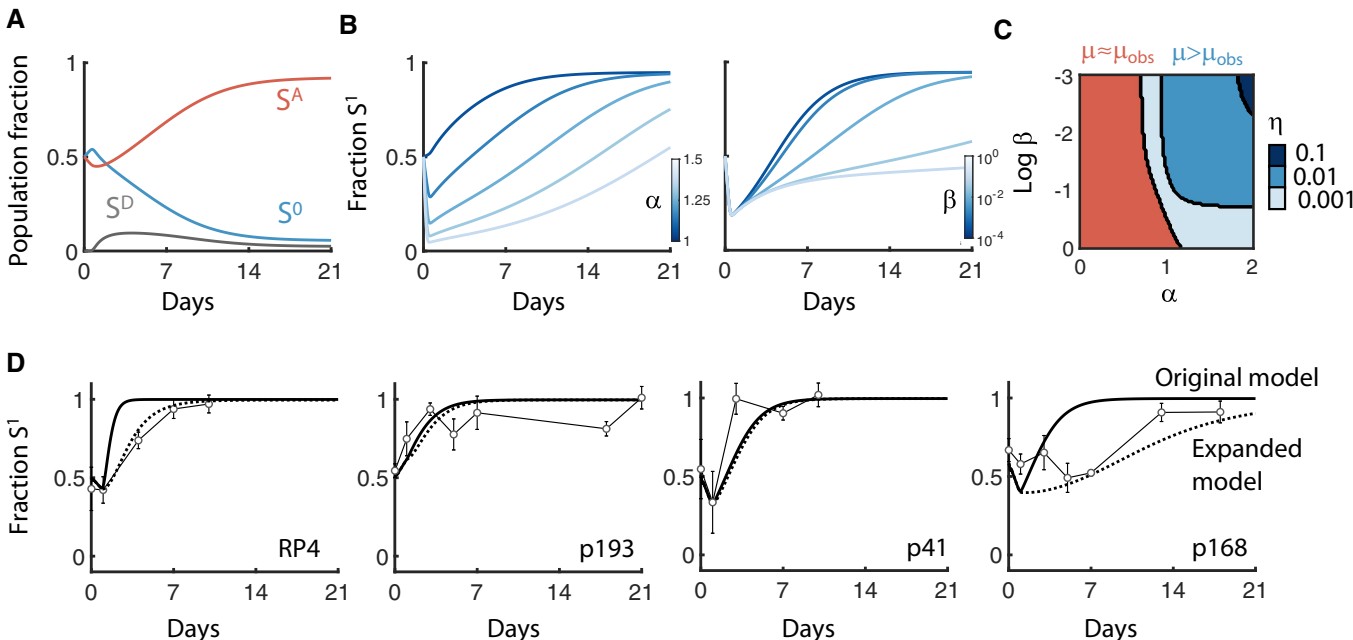

**Figure 5. Fitness cost versus acquisition cost.**

A   Long-term temporal dynamics of $S^A$, $S^O$, and $S^D$ are shown in red, blue, and gray, respectively, from the main model (Fig 4A). X-axis is time over 21 days, and y-axis is the fraction of each population.

B   A population is initiated with a 1:1 ratio of $S^O$ and $S^A$ and the total plasmid-carrying population fraction ($S^1 = S^A + S^D$) is tracked over time. *Left*: $\beta$ is held constant ($\beta = 0.01$) and $\alpha$ is increased from no cost ($\alpha = 1$) to high cost ($\alpha = 1.5$). *Right*: $\alpha$ is held constant ($\alpha = 1.2$) and $\beta$ is increased from slow transition ($\beta = 10^{-4}$) to rapid transition ($\beta = 1$).

C   Heat map shows where the observed growth rate of $S^1$ ($\mu_{obs}$, calculated using Appendix equation S6) differed from the maximum growth rate under ideal conditions (e.g., $\mu$, if there is no acquisition cost). A 98% threshold was used to numerically define the region where $\mu_{obs}$ differed significantly from $\mu$ (e.g., $\mu_{obs}/\mu < 0.98$). Any $\alpha$ and $\beta$ combination meeting this criterion is colored blue and are red otherwise. Changing this threshold did not qualitatively change conclusions (Appendix Fig S9). Changing the conjugation efficiency ($\eta$) shifts the boundary (increasing from light to dark shades of blue).

D   Validation of modeling predictions using four plasmids from left to right: RP4 (in this study), p193, p41, and p168 (from previous work). Data are reproduced with permission from Nature Communications (Lopatkin *et al*, 2017), under the Creative Commons Attribution 4.0 International License. Marker shapes and colors were modified for visualization purposes. Solid line shows original model fit (e.g., Appendix equation S1–S2). Dotted lines show updated model fit (e.g., Appendix equations S3–S5). Experiments were performed at least twice. Error bars represent the standard deviation of four to six measurements.

# Discussion

In this study, we experimentally and computationally separated the acquisition and fitness consequences of plasmids following conjugation and then quantified their relative impacts on both short- and long-term growth dynamics. Conjugation dynamics are traditionally described using a combination of conjugation efficiency estimates (i.e., how fast the plasmid transfers) and relative growth rates (i.e., the selection dynamics of resulting transconjugants) (Stewart & Levin, 1977; Simonsen, 1991). Overall, we observed that *de novo* transconjugants exhibit a transient growth defect that is recovered within 24 h. This plasmid acquisition cost appeared to be general for diverse well-characterized and clinically relevant plasmid types, as well as experimental conditions. We also demonstrate that the acquisition cost is general to four incompatibility groups. Tellingly, all incF plasmids studied here were uniquely found to impose no measurable acquisition cost. Given that incompatibility groups employ different replication/partitioning mechanisms, often in conjunction with specific host gene expression, it can be reasonably assumed that the acquisition cost may vary among plasmid-host combinations. However, we show that acquisition costs arise in different recipient strains *and* different recipient species, further demonstrating the generality and scope of this phenomenon.

Our results update our canonical interpretation of growth dynamics during conjugation in that they may not be fully defined by a static fitness cost, which represents a third critical and currently underappreciated component. Indeed, when acquisition costs are particularly large, disregarding their effect can result in inaccurate persistence predictions (Appendix Fig S8), which is often a key objective in modeling *in vivo* microbial communities and/or microbial risk assessment.

Growth effects of conjugal plasmids are often represented by easily measured transconjugant growth rates or competition experiments. However, we show that the acquisition of a new plasmid forces the host cell to adapt over shorter timescales, placing an additional burden that modulates growth phenotypes. Given the dual antibiotic selection in our experimental setup, we note that observed acquisition costs may arise due to a delay in corresponding resistance gene expression. However, we identified several cases in which no acquisition costs were observed despite transconjugants being subject to identical dual antibiotic selection, e.g., low metabolic efficiency conditions and for various incF

plasmids. Moreover, acquisition costs were variable within specific dual antibiotic combinations. These results suggest that, although resistance expression may contribute to acquisition costs, it is not the primary determinant.

Using simulations, we also show that this time-dependent growth rate is dictated by both the growth of newly formed transconjugants, as well as their transition to a fully adapted phenotype. In general, minimal growth rate depression and rapid transition together increase the likelihood that a plasmid will be maximized in a given heterogeneous environment. Importantly, these results reveal that the window immediately following plasmid acquisition may represent a critical time interval in which a cell must adjust to increasing metabolic demand in order to survive.

Here, phenotypic adaptation and selection are based on pre-existing heterogeneity. Though our experimental methods minimize the likelihood of genetic mutation, we note that this possibility does remain; nonetheless, our model fully accounts for the observed dynamics, suggesting that mutations are unlikely, or at minimum unnecessary, to explain these phenomena. Moreover, the recovered growth rate did not negate the overall fitness cost, which is the primary indicator of compensatory mutations observed in previous studies (Dahlberg & Chao, 2003). Although compensatory mutations that ameliorate fitness cost can sometimes occur as rapidly as several days, they typically emerge over hundreds of generations. Therefore, it remains to be seen whether mutations that mitigate fitness costs affect acquisition costs as well.

Critically, our model suggests that competition plays a key role in dictating plasmid dynamics. Specifically, plasmid acquisition costs can lead to delays in maximum transconjugant growth. In a complex microbial community, this delay may or may not be advantageous—it might represent a transiently available niche that could be filled by a competitor strain, or a window for newly formed populations to optimize their metabolic adaption. These possibilities have several implications in the fields of antibiotic therapy and engineered microbial communities. For example, conjugation inhibitors have been proposed as a potential adjuvant for antibiotic treatment (Warnes *et al*, 2012). Our results suggest that analogous adjuvants that prolong the cellular adaptation process (e.g., by magnifying the metabolic burden/effects of a plasmid) may also mitigate the spread of plasmids in heterogeneous populations. From a synthetic standpoint, our results suggest that accounting for acquisition cost may be essential to accurately predict complex heterogeneous dynamics, particularly in *in vivo* settings where plasmids may be incompletely characterized.

Overall, this analysis demonstrates that, in addition to fitness costs, plasmid acquisition costs are an additional critical component of temporal conjugation dynamics. Moreover, acquisition costs should be included in strain and plasmid-level characterization to facilitate accurate consortium predictions and design. As such, this study represents a complementary perspective in the study of HGT and microbial community dynamics: How a cell pays the immediate cost of acquiring genetic material may dictate the extent of the benefit it derives thereof.

# Materials and Methods

**Reagents and Tools Table**

| Reagent/Resource | Reference or Source | Identifier/ Catalog No |
|---|---|---|
| **Media** | | |
| Luria-Bertani (LB) | BD Difco from Fisher Scientific | DF0446 07 5 |
| M9CA | Amresco M9CA medium broth powder | lot no. 2055C146 |
| **Plasmids** | | |
| *Donors and recipients* | | |
| *Strain Genotype* | | |
| RP4 donor | (Lopatkin *et al*, 2016) | DA32838 Eco galK::cat-J23101-dTomato |
| pR donor | (Lopatkin *et al*, 2016) | DA26735 Eco lacIZYA::FRT, galK::mTagBFP2-amp |
| p41 donor | (Lopatkin *et al*, 2016) | *E. coli* isolate number 41 |
| p168 donor | (Lopatkin *et al*, 2016) | *E. coli* isolate number 168 |
| p193 donor | (Lopatkin *et al*, 2016) | *E. coli* isolate number 193 |
| p283 donor | (Händel *et al*, 2015) | ESBL 242 |
| R6K donor | (Lopatkin *et al*, 2016) | *E. coli* C600 |
| R6Kdrd donor | Generous gift from D. Mazel (Baharoglu *et al*, 2010) | *E. coli* Dh5a |
| R1 donor | Generous gift from F. Dionisio and J. Alves Gama (Gama *et al*, 2020) | *E. coli* MG1655 Dara |
| R1drd donor | Generous gift from F. Dionisio and J. Alves Gama (Gama *et al*, 2020) | *E. coli* MG1655 Dara |
| pRK100 donor | Generous gift from T. Sysoeva | *E. coli* HB101 |

**Reagents and Tools table**   (continued)

| Reagent/Resource | Reference or Source | Identifier/ Catalog No |
|---|---|---|
| RIP113 donor | Generous gift from D. Mazel (Baharoglu et al, 2010) | E. coli Dh5a |
| RB933 recipient | Generous gift from I. Gordo (Leónidas Cardoso et al, 2020) | E. coli lacIZYA::scar galK::cat-YFP ΔgatZ::FRT-aph-FRT rpoB$^{H526Y}$ |
| MG1655 | (Lopatkin et al, 2016) | E. coli MG1655 (K-12 F⁻ λ⁻ ilvG⁻ rfb-50 rph-1) |
| DA838F | (Lopatkin et al, 2016) | DA32838 Eco galK::cat-J23101-dTomato |
| DA838 | (Lopatkin et al, 2016) | DA32838 Eco galK::cat-J23101-dTomato |
| P (recipient for p41, p193, and p168) | This study, lab stock | E. coli MG1655 (K-12 F⁻ λ⁻ ilvG⁻ rfb-50 rph-1) |
| KPN recipient | (Gomez-Simmonds et al, 2015) | Klebsiella pnseumoniae isolate KP0064, ST17 |
| **Software** | | |
| MATLAB v. R2020a | https://www.mathworks.com/products/matlab.html | N/A |
| **Tools** | | |
| Tecan infinite Mplex plate reader | Tecan | N/A |

## Methods and Protocols

### Strains, media, and growth conditions

Experiments were initiated with single clones picked from agar plates, inoculated in 2 ml Luria-Bertani (LB) media, and incubated overnight at 37°C for exactly 16 h while shaking at 250 rpm. When applicable, LB media was supplemented with specific antibiotics. For example, to measure the acquisition cost of RP4, D, R, and adapted T were grown with 50 μg/ml kanamycin (Kan), 50 μg/ml spectinomycin (Spec), or both, respectively (Appendix Table S1A). For all other donor and recipient combinations, the respective antibiotics and concentrations can be found in Appendix Table S1B. All experiments were performed in M9 medium (M9CA medium broth powder from Amresco, lot no. 2055C146, containing 2 mg/ml casamino acid, supplemented with 2 mM MgSO$_4$, 0.1 mM CaCl$_2$ and 0.4 % w/v glucose.)

### Generating adapted RP4 transconjugants

To generate adapted T, individual clones of D and R were grown overnight according to the previous description. After 16 hours, all cultures were resuspended 1:1 in M9CA. Equal volumes (400 μl each) of D and R were mixed in an Eppendorf tube and incubated for 1 h in a cooling incubator at 25°C. Following the conjugation period, mixtures were streaked onto Spec-Kan plates and grown for 16 h at 37°C; this is sufficiently long to enable physiological adaptation (Erickson et al, 2017) without genetic mutations (Harrison et al, 2016; Lopatkin et al, 2017) and is consistent with previous protocols for establishing transconjugants for fitness cost estimates (Buckner et al, 2018; Dimitriu et al, 2019).

### Quantifying the acquisition cost for RP4

To generate de novo T, individual clones of D and R were grown overnight and conjugated as described above. In parallel, adapted T colonies were grown overnight to establish the standard curve. During the conjugation period of D and R, 800 μl of adapted T were also aliquoted into an Eppendorf tube and placed in the 25°C cooling incubator. Following the conjugation period, de novo T was quantified from the mixture of D and R using either colony forming units (CFU) or via dilution into the plate reader. For CFU, the D and R mixture was serially diluted in 10-fold dilution increments. 10 μl of all dilution factors ($10^0$–$10^7$) were spotted onto Spec-Kan agar plates, then incubated for 16 h at 37°C and counted. For the plate reader, the D and R mixture was diluted into M9CA media containing Spec-Kan, and 200 μl were aliquoted into wells of a 96-well plate in technical triplicates. As determined by pilot experiments, de novo T was diluted at 1,000× to result in approximately 2,000 cells per well, which was found to be low enough to prevent background growth or conjugation.

CFU of adapted T was also quantified at this time using the same protocol as described above. The standard curve was generated using seven dilution factors of adapted T (Fig 2B). Adapted T cells were diluted in M9CA medium with Spec-Kan, and then, 200 μl aliquots of each dilution factor were plated on the same 96-well plate, also in technical triplicate. All wells were then covered in 50 μl of mineral oil to prevent evaporation and immediately placed into a Tecan plate reader. In all cases, absorbance readings at 600 nm were taken every 15 min for at least 24 h until all cells reached stationary phase. All RP4 data were conducted in at least biological triplicates.

### Generality of acquisition costs with other plasmids

The same conjugation protocol described above was used for all additional plasmids with few modifications. First, the antibiotics used to select for transconjugants in each case depended on the plasmid-encoded resistance genes and the compatible recipient strain (Appendix Table S1B). Also, out of the seven dilution factors used for the RP4 and pR standard curves, we found the subset of dilution factors at $10^2$X, $10^4$X, and $10^6$X sufficient to accurately quantify both plasmids' acquisition costs (as when determined with all dilutions). Thus, these three dilution factors were used to generate standard curves for the remainder of the experiments. The maximum cell density (defined by OD$_{600}$) and growth rates differed depending on the strain and plasmid, based on conjugation efficiency. In all cases, the dilution factor into the plate reader was determined based on pilot experiments to identify the highest initial density in the wells without observable background conjugation. In cases where background conjugation could not be eliminated

entirely, growth was subtracted such that the exponential phase remained. To remain consistently within exponential phase, the threshold cell density was set to be equal to 50% of the maximum density achieved for each experiment; this value appeared to most consistently select the region indicated in Fig 2A as described in the Results. Altering the threshold within the exponential growth phase does not qualitatively impact our conclusions. Finally, we note that acquisition costs were highly reproducible between biological replicates (see Appendix Fig S10A); in many cases, variability was greater among individual wells on a given day rather than between intra-day mean values, likely due to the low number of cells per well once diluted into 96-well plates. Where biological duplicate results were reproducible and did not lead to different statistical conclusions, we instead focused on technical variability within a representative replicate. Therefore, by utilizing the most variable data in any given case, we maximize the rigor of associated statistical conclusions. In each case, the replicate types used to generate statistics are shown in Appendix Table S3A (Fig 3H). We note that regardless of the replicates used to generate statistics, specific acquisition costs remain identical. Furthermore, the relationship between acquisition and fitness cost did not change depending on whether biological replicate averages were used instead (Appendix Fig S10B).

### Calculating growth rate

To calculate the growth rates, $OD_{600}$ growth curves were fit using the Baranyi equation: $y = y_0 + \mu A(t) - \ln\left(1 + \frac{e^{\mu A(t)} - 1}{e^C}\right)$, and $A(t) = t + \frac{1}{\mu}\ln\left(e^{-\mu t} + e^{-\mu \lambda} - e^{\mu(t+\lambda)}\right)$. Here, $y$ is the log-transformed fold change in cell density (e.g., $\log(N/N_0)$), C is the increase in cell density between $y_0$ and maximum cell density, $\mu$ is the maximal growth rate, $y_0$ is the initial cell density, and $\lambda$ is the lag time (Zwietering et al, 1990). The maximal growth rate was taken exactly as the growth rate, in units hour$^{-1}$. To validate this growth metric, three additional calculation methods were used including the modified Gompertz equation, $y = Ae^{-e^{\frac{\mu e}{A}(\lambda - t) + 2}}$, and the modified Logistic equation, $y = \frac{A}{\left[1 + e^{\frac{4\mu}{A}(\lambda - t) + 2}\right]}$ (Zwietering et al, 1990). Custom MATLAB scripts were written to fit the growth curves using each method by implementing non-linear least-squares fitting. Geometric lag time estimation was determined by locating the time corresponding to the maximum growth rate, and extending a tangent line using the two points before and after $t$. $\lambda$ was calculated to be equal to the x-intercept of this tangent line. In addition to these three established growth estimation methods, a four method was employed (which we refer to as Prensky) to quantify and compare maximal growth rates. The Prensky method was designed using curve-smoothing and numerical differentiation to identify the time $t$, at which the maximum derivative occurred within the region of exponential growth. The maximal growth rate was then calculated as the linear slope of the tangent line passing between the two time points above and below the t (equation 4):

$$\mu = \frac{((t+2) - (t-2))}{y_2 - y_1} \quad (4)$$

where $y_2 > y_1$, where $y_2$ and $y_1$ are the $OD_{600}$ at times $t+2$ and $t-2$, respectively. The Prensky lag time was found by the x-intercept of

the tangent line passing through the maximum growth rate, consistent with the geometric lag time in Fig 2A.

### Competition experiments

All competition experiments were performed in a previous publication and the data were reproduced with permission from Nature Communications (Lopatkin et al, 2017), which is licensed under the Creative Commons Attribution 4.0 International License (http://creativecommons.org/licenses/by/4.0/). Briefly, plasmid-free and plasmid-carrying populations were mixed in a 1:1 ratio and grown over successive generations for between 14-21 days. Every 24 h, the populations were diluted 10,000×, and CFU was monitored at regular intervals on double-selective agar.

### Model calculations

Simulations were run to calculate the observed growth rate, $\mu_{obs}$, and the maximum specific growth rate, $\mu$, for a range of $\alpha$ and $\beta$ values (Fig 4A). $\mu_{obs}$ was calculated as described in Appendix equation S6. $\mu_{obs}$ was determined to be numerically different from $\mu$ based on any observed growth rate that fell below 98% of the maximum value (e.g., if $\mu_{obs} < 0.98*\mu$). In all cases for data fitting, was calculated using the fminsearch function in MATLAB. fminsearch is an optimization function that minimizes any user-specified objective function, and requires an initial estimate of the parameter(s) to be fitted. In our case, we implemented this fitting by minimizing the difference between the ODE solution of $S^D + S^A$, and the raw data curves as shown in Appendix Fig S7; since $\rho$ was constrained by experimentally estimating the ratio of growth rates between adapted and de novo populations, $\beta$ was the only remaining free parameter to be fitted. Thus, for each plasmid, the initial $\beta$ estimate was set to be the geometric lag time of the corresponding growth curve, and the output of the fminsearch optimization was an estimated $\beta$ that best fit our experimental data.

## Data availability

All raw cost data and sample analysis codes are available on the laboratories github: https://github.com/ajlopatkin/acquisition-cost-growthrate.

Expanded View for this article is available online.

### Acknowledgements

We thank T. Sysoeva for the pRK100 plasmid, J. P. Alves and F. Dionisio for the R1 and R1drd plasmids, as well as D. Mazel for the R6kdrd and RIP113 plasmids. We also thank I. Gordo for the RB933 Rif$^R$ recipient strain.

### Author contributions

AJL conceived the research, designed and conducted experiments, performed all modeling, interpreted the data, and wrote the manuscript. HP assisted in experiments, data analysis, and manuscript editing. AG-S and A-CU provided the KPN strain, and assisted in manuscript editing.

### Conflict of interest

The authors declare that they have no conflict of interest.

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
