## [Review Process File · Molecular Systems Biology]

Conjugation dynamics depend on both the plasmid acquisition cost and the fitness cost

Hannah Prensky, Anne-Catrin Uhlemann, Angela Gomez-Simmonds, and Allison Lopatkin

DOI: [10.15252/msb.20209913](https://doi.org/10.15252/msb.20209913)

Corresponding authors: Allison Lopatkin (alopatkin@barnard.edu)

Review Timeline:

Submission Date:	8th Aug 20
Editorial Decision:	8th Sep 20
Revision Received:	6th Dec 20
Editorial Decision:	14th Dec 20
Revision Received:	11th Jan 21
Accepted:	13th Jan 21

Editor: Maria Polychronidou

Transaction Report:

8th Sep 2020

Manuscript Number: MSB-20-9913, Conjugation dynamics depend on both the plasmid acquisition cost and the fitness cost

Dear Dr. Lopatkin,

Thank you again for submitting your work to Molecular Systems Biology. We have now heard back from the two referees who agreed to evaluate your study. As you will see below, the reviewers acknowledge that the presented findings seem interesting. However, they raise a series of concerns, which we would ask you to address in a major revision.

Without repeating all the issues listed below, one of the more fundamental issues is raised by reviewer #1, who points out that as it stands the study remains somewhat preliminary for a broad-audience journal. This referee recommends expanding the study by either providing further evidence for the generality of the presented findings, providing more concrete insights into the molecular mechanisms (the referee makes constructive suggestions in this regard), or into the consequences for conjugation dynamics. We would therefore ask you to include follow up analyses in one of these directions in order to strengthen the study. Reviewer #1 provides some suggestions on analyses that could potentially be performed to derive further mechanistic insights but we would be open to discussing other types of analyses that could be included instead. Please let me know in case you would like to discuss in further detail any of the issues raised. All issues raised by the referees would need to be satisfactorily addressed.

On a more editorial level, we would ask you to address the following issues:

- Please include 5 keywords.
- Please provide a .doc version of the manuscript text (including legends for main figures and tables) and individual production-quality files for the main figures.
- A Conflict of Interest statement should be included in the main text.
- We have replaced Supplementary Information by the Expanded View (EV format). In this case, all additional figures can be included in a PDF called Appendix. Appendix figures and Tables should be labeled and called out as: "Appendix Figure S1, Appendix Figure S2... Appendix Table S1..." etc. Each legend should be below the corresponding Figure/Table in the Appendix. Please include a Table of Contents in the beginning of the Appendix. For detailed instructions regarding expanded view please refer to our Author Guidelines: .
- Please provide a "standfirst text" summarizing the study in one or two sentences (approximately 250 characters), three to four "bullet points" highlighting the main findings and a "synopsis image" (550px width and max 400px height, jpeg format) to highlight the paper on our homepage.
- All Materials and Methods need to be described in the main text. We would encourage you to use 'Structured Methods', our new Materials and Methods format. According to this format, the Material and Methods section should include a Reagents and Tools Table (listing key reagents,

experimental models, software and relevant equipment and including their sources and relevant identifiers) followed by a Methods and Protocols section in which we encourage the authors to describe their methods using a step-by-step protocol format with bullet points, to facilitate the adoption of the methodologies across labs. More information on how to adhere to this format as well as downloadable templates (.doc or .xls) for the Reagents and Tools Table can be found in our author guidelines: . An example of a Method paper with Structured Methods can be found here: .

- The code and data need to be made publicly available. Please use the Data availability section to describe how the data and code have been made available. This section needs to be formatted according to the example below:

The datasets and computer code produced in this study are available in the following databases:

- Chip-Seq data: Gene Expression Omnibus GSE46748

(<https://www.ncbi.nlm.nih.gov/geo/query/acc.cgi?acc=GSE46748>)

- Modeling computer scripts: GitHub (<https://github.com/SysBioChalmers/GECKO/releases/tag/v1.0>)

- [data type]: [full name of the resource] [accession number/identifier] ([doi or URL or identifiers.org/DATABASE:ACCESSION])

- Due to the quantitative nature of the study we would encourage you to provide the Source Data for the Figure panels showing essential quantitative information. Source Data for main figures should be provided in .zip Folders labeled "Source data for Figure X". Please provide one .zip folder for each of the main figures. Source Data for Appendix Figures should all be provided in one single .zip folder labeled "Source Data for Appendix". Further information regarding Source Data can be found here: .

- For data quantification: please specify the name of the statistical test used to generate error bars and P values, the number (n) of independent experiments (specify technical or biological replicates) underlying each data point and the test used to calculate p-values in each figure legend. The figure legends should contain a basic description of n, P and the test applied. Graphs must include a description of the bars and the error bars (s.d., s.e.m.).

- Please note that our editorial policy does not allow "Data not shown".

- The References need to be formatted according to the Molecular Systems Biology reference style.

- When you resubmit your manuscript, please download our CHECKLIST

(<http://bit.ly/EMBOPressAuthorChecklist>) and include the completed form in your submission.

Please note that the Author Checklist will be published alongside the paper as part of the transparent process

(<https://www.embopress.org/page/journal/17444292/authorguide#transparentprocess>).

If you feel you can satisfactorily deal with these points and those listed by the referees, you may wish to submit a revised version of your manuscript. Please attach a covering letter giving details of the way in which you have handled each of the points raised by the referees. A revised manuscript will be once again subject to review and you probably understand that we can give you no guarantee at this stage that the eventual outcome will be favorable.

Kind regards,

Maria

Maria Polychronidou, PhD
Senior Editor
Molecular Systems Biology

If you do choose to resubmit, please click on the link below to submit the revision online *within 90 days*.

Link Not Available

IMPORTANT: When you send your revision, we will require the following items:

1. the manuscript text in LaTeX, RTF or MS Word format
2. a letter with a detailed description of the changes made in response to the referees. Please specify clearly the exact places in the text (pages and paragraphs) where each change has been made in response to each specific comment given
3. three to four 'bullet points' highlighting the main findings of your study
4. a short 'blurb' text summarizing in two sentences the study (max. 250 characters)
5. a 'thumbnail image' (550px width and max 400px height, Illustrator, PowerPoint or jpeg format), which can be used as 'visual title' for the synopsis section of your paper.
6. Please include an author contributions statement after the Acknowledgements section (see <https://www.embopress.org/page/journal/17444292/authorguide>)
7. Please complete the CHECKLIST available at (<http://bit.ly/EMBOPressAuthorChecklist>). Please note that the Author Checklist will be published alongside the paper as part of the transparent process (<https://www.embopress.org/page/journal/17444292/authorguide#transparentprocess>).
8. Please note that corresponding authors are required to supply an ORCID ID for their name upon submission of a revised manuscript (EMBO Press signed a joint statement to encourage ORCID adoption). (<https://www.embopress.org/page/journal/17444292/authorguide#editorialprocess>)

Currently, our records indicate that the ORCID for your account is 0000-0003-0018-9205.

Link Not Available

The system will prompt you to fill in your funding and payment information. This will allow Wiley to send you a quote for the article processing charge (APC) in case of acceptance. This quote takes into account any reduction or fee waivers that you may be eligible for. Authors do not need to pay any fees before their manuscript is accepted and transferred to the publisher.

***** PLEASE NOTE ***** As part of the EMBO Press transparent editorial process initiative (see our Editorial at <http://dx.doi.org/10.1038/msb.2010.72>), Molecular Systems Biology publishes online a Review Process File with each accepted manuscripts. This file will be published in conjunction with your paper and will include the anonymous referee reports, your point-by-point response and all pertinent correspondence relating to the manuscript. If you do NOT want this File to be published,

please inform the editorial office at msb@embo.org within 14 days upon receipt of the present letter.

Reviewer #1:

This manuscript describes and quantifies a phenomenon that has been rarely explored previously: a so-called acquisition cost for conjugative plasmids, in which plasmid cost shortly after acquisition is larger than subsequent cost in established transconjugants. The results presented here use mostly *E. coli* and the conjugative plasmid RP4, with some measures using other plasmids or host strains to demonstrate generality. The cost is quantified with optical density measurements, comparing recent transconjugants with established ones. The authors further model the consequences of this acquisition cost on plasmid population dynamics and compare modelling results with their previous data.

The manuscript combines several types of results. The first is a technique to evaluate acquisition cost by optical density measurements. The authors demonstrate the validity of their method with appropriate and convincing control experiments. Next, they demonstrate the existence of an acquisition cost focusing on RP4 plasmid and then explore the generality and consequences of this phenomenon. Acquisition costs have not been the primary focus of many studies yet, and the results presented here are interesting and well demonstrated. However, some similar results have been obtained previously, to my knowledge at least in Fernandez Lopez et al, PLoS Genetics 2014, (a paper cited by the authors). In that paper, the behaviour of R388 plasmid, another broad-host-range plasmid, was analysed, showing among other things that transconjugant growth rate immediately after conjugation is strongly decreased, but that fact is not mentioned here. The behaviour of RP4 is well analysed, and a few other plasmids are measured too, some at two temperatures. This allows the authors to generalize their observations to more than one plasmid, however not enough plasmids/conditions are tested to really explore in depth the generality of this result or provide insight into what the mechanisms responsible might be. The authors mention metabolic perturbation as a possible cause, but do not address it directly. Finally, the consequences of the acquisition cost on plasmid population dynamics are explored with modelling, describing in which scenarios the acquisition cost most limits plasmid spread. However, I suspect the predictions concerning plasmid prevalence would be altered significantly if considering transfer shortly after acquisition (see below).

Overall, the acquisition cost described here is a very interesting physiological observation, but I am not convinced it will be of interest to a wide readership without more in-depth characterization of at least one of the aspects covered here (generality of the phenomenon, molecular mechanisms, or consequences on conjugation dynamics).

Major points

- The authors explanation about the mechanistic reason for acquisition costs was not clear to me. The authors mention 'metabolic perturbation' as a cause, and claim to demonstrate its importance (end of introduction), but do not discuss the molecular processes involved, or provide data to support this. References for metabolic effects or physiological adaptation are cited but the concepts need explained within the manuscript, and possibly investigated in more detail. Would for instance expression of plasmid genes be included or do the authors have more specific metabolic effects in mind? The 'longer replication time' of RP4 (page 7) is also proposed as a reason for its

high acquisition cost -why would it affect acquisition cost instead of the fitness cost?

Increased expression upon first entry in the cell, and possible reasons for this could be elaborated on. Specifically, one phenomenon not discussed here is transitory derepression of plasmid conjugative transfer genes, which leads to high rates of transfer after first entering the cell. Transitory derepression and its consequences for plasmid and host fitness have been modelled previously (Lundquist and Levin, *Genetics* 1986; Dionisio, *Evol Ecol Research* 2005), as a strategy for plasmid epidemic spread while minimizing costs. Importantly, it makes plasmids highly transferable for the first generations after transfer, making the assumption that conjugation from SD is negligible in the model inappropriate. This will likely affect the consequences on plasmid population dynamics.

Some mechanistic data could help provide some insight into what is responsible for this acquisition cost. Transcriptomics comparing RP4 recent and established transconjugants could pinpoint the plasmid or host genes involved; plasmids with derepressed conjugation are also available, and could maybe be used here: for instance, F plasmid or R1 derepressed mutants (see Haft et al, *ISME* 2009) might have a high fitness cost that does not disappear, whereas repressed versions could lead to a high acquisition cost but lower fitness one.

Overall, I think some restructuring of the data could make the main points of the paper more convincing, even without additional data. Some of the interesting data are actually a bit buried under many controls, or not exploited as much as they could, appearing more anecdotal than they deserve. An extreme example is the effect of recipient strain. Having data on *Klebsiella* is great and indeed generalises the results, but then the BW25113 strain is really not needed (at least not in the main figure) as it is almost identical to MG1655. In Figure 3, the effects of strain BW25113 and dilution on acquisition cost could be moved to supplementary results, or at least combined in a single graph. On the other hand, figure 3F is very interesting and should be given more space and discussion!

The existing data could also be exploited more to obtain more quantitative results. For instance, could the authors estimate the number of generations during which the acquisition cost is present from their results?

- I have one main methodological issue: could some of the delay in growth be due specifically to the requirement for expression of plasmid-carried resistance genes with antibiotic selection? In that case this would not strictly represent a cost of plasmid acquisition but a delay in plasmid benefits, and could be seen as an artefact of the method used here.

Minor points

- There is a problem with reference numbering, with several references duplicated.
- page 2 definitions: strictly speaking, cell-to-cell contact would include other means of HGT like nanotubes, not only conjugation. Similarly, the cell fusion shown in Figure 1A drawing is confusing when representing conjugation.
- page 5: 'For each panel ...' should go in the figure legend.
- page 7 'mobile plasmids' are mobilisable plasmids?
- page 9: ref 27 is not about a model of conjugation.
- page 14: from 'more generally', the end of results might be more suited to the discussion
- there are several minor mistakes in the writing, of which I have probably missed some: for instance e.g. is used for referring to explanations not examples; page 22 'the prensky method ...' sentence

lacks a verb.

- page 23: I am no expert in this but more details should probably be given on the model fitting details, than 'using an optimisation function'- I would not know how to replicate this from these methods.

Across the manuscript, full p-values should be stated (not only < 0.05).

- page 6 (and all related figures): why not use the time to threshold itself as a measure of acquisition cost, instead of predicted cell densities? It would seem more intuitive to me, a higher cost directly leading to a longer time to grow instead on calculating the discrepancy between predicted and actual cell densities for initial transconjugants (which are not the factor actually modified here)

- do the authors have a hypothesis about why there is high variation in individual replicates in Figure 1C? Are the recently acquired transconjugants always more variable than the adapted ones?

- Figure S6B could be plotted in log scale as well, as the initial dynamics is interesting and cannot be seen here.

Reviewer #2:

This is a clear and interesting paper reporting "acquisition costs" of plasmids, that are neatly distinguished from fitness costs. The paper combines well-designed and controlled experiments with mathematical modelling to develop a full description of acquisition costs for a focal plasmid-host association and then expands this to other plasmid-host associations to prove the generality of the phenomenon. Finally, they show that a new augmented model including acquisition costs and fitness costs better explains longer term plasmid dynamics especially where acquisition costs are appreciable.

This is a well-written and presented study. The paper is very clear and easy for follow and the work builds logically upon the previous work by the group. The discovery of acquisition costs is novel and interesting, and the inclusion of multiple plasmid-host associations reveals some very intriguing variability in the magnitude of acquisition costs.

Given the high quality of the work and its presentation I do not have very many critical comments. I think the discussion would benefit from a few words speculating why acquisition costs might vary among plasmid-host associations. Another potentially interesting subject for discussion may be whether acquisition costs would be under selection e.g. through plasmid evolution to minimise these. Also, a key missing reference that would add to the discussion of compensatory evolution dynamics is [Hall et al 2020 Microbiology] which shows the potential for very rapid compensatory evolution.

Reviewer #1:

This manuscript describes and quantifies a phenomenon that has been rarely explored previously: a so-called acquisition cost for conjugative plasmids, in which plasmid cost shortly after acquisition is larger than subsequent cost in established transconjugants. The results presented here use mostly *E. coli* and the conjugative plasmid RP4, with some measures using other plasmids or host strains to demonstrate generality. The cost is quantified with optical density measurements, comparing recent transconjugants with established ones. The authors further model the consequences of this acquisition cost on plasmid population dynamics and compare modelling results with their previous data.

The manuscript combines several types of results. The first is a technique to evaluate acquisition cost by optical density measurements. The authors demonstrate the validity of their method with appropriate and convincing control experiments. Next, they demonstrate the existence of an acquisition cost focusing on RP4 plasmid and then explore the generality and consequences of this phenomenon. Acquisition costs have not been the primary focus of many studies yet, and the results presented here are interesting and well demonstrated. However, some similar results have been obtained previously, to my knowledge at least in Fernandez Lopez et al, PLoS Genetics 2014, (a paper cited by the authors). In that paper, the behaviour of R388 plasmid, another broad-host-range plasmid, was analysed, showing among other things that transconjugant growth rate immediately after conjugation is strongly decreased, but that fact is not mentioned here. The behaviour of RP4 is well analysed, and a few other plasmids are measured too, some at two temperatures. This allows the authors to generalize their observations to more than one plasmid, however not enough plasmids/conditions are tested to really explore in depth the generality of this result or provide insight into what the mechanisms responsible might be. The authors mention metabolic perturbation as a possible cause, but do not address it directly. Finally, the consequences of the acquisition cost on plasmid population dynamics are explored with modelling, describing in which scenarios the acquisition cost most limits plasmid spread. However, I suspect the predictions concerning plasmid prevalence would be altered significantly if considering transfer shortly after acquisition (see below).

Overall, the acquisition cost described here is a very interesting physiological observation, but I am not convinced it will be of interest to a wide readership without more in-depth characterization of at least one of the aspects covered here (generality of the phenomenon, molecular mechanisms, or consequences on conjugation dynamics).

We are encouraged that the Reviewer is overall enthusiastic about the topic and significance of our manuscript. We are grateful for the feedback, and have fully addressed all of the specific points raised below, and revised the main text accordingly. In particular, we have taken all three major recommendations into consideration and expanded our results in terms of generality, mechanism, and model exploration. We did this by studying seven additional plasmids, including those enabling us to draw mechanistic insights, measuring acquisition cost under additional growth conditions, and adding specific mechanistic hypotheses to our computational simulations. Combined, our study now measures acquisition costs for 12 diverse laboratory and clinical plasmids, spanning five incompatibility groups, recipient strains and species, and under various growth conditions, resulting in almost 30 different plasmid/strain/environmental combinations.

To briefly summarize our revisions, we quantified the acquisition cost for the additional plasmids R1, R1drd, R6K, R6Kdrd, p283, and RIP113, and pRK100. Overall, we detected acquisition costs for 9 of the total 12 plasmids studied; all plasmids that exhibited an acquisition cost in one environment also exhibited acquisition costs in separate environments, establishing the robustness of these results. Interestingly, the three plasmids that did not impose acquisition costs all belong to incompatibility group F, indicating inc- to inc-group variability that may arise from differences in replication control. Moreover, to improve our understanding of acquisition cost's generality, we performed additional experiments under various growth conditions and showed that the acquisition cost increases as cell growth becomes more inefficient. To provide further insights into molecular mechanisms, we compared the acquisition cost of naturally repressed plasmids with their mutant derepressed counterparts, namely, R1/R1drd and R6K/R6Kdrd. We did not observe qualitative

differences in the acquisition costs imposed between each pair. Finally, we incorporated plasmid transfer shortly after acquisition in our model simulations; this did not significantly influence model predictions or final conclusions. These and additional experiments/analyses are discussed in full detail below.

Major points:

1. The authors explanation about the mechanistic reason for acquisition costs was not clear to me. The authors mention 'metabolic perturbation' as a cause, and claim to demonstrate its importance (end of introduction), but do not discuss the molecular processes involved, or provide data to support this. References for metabolic effects or physiological adaptation are cited but the concepts need explained within the manuscript, and possibly investigated in more detail. Would for instance expression of plasmid genes be included or do the authors have more specific metabolic effects in mind? The 'longer replication time' of RP4 (page 7) is also proposed as a reason for its high acquisition cost -why would it affect acquisition cost instead of the fitness cost?

We agree with the reviewer that our original version did not sufficiently describe in detail the mechanisms underlying the acquisition cost. To address this, we conducted additional experiments that support our hypothesis that the acquisition cost is reflective of cells responding to increased metabolic demand immediately after conjugation. Specifically, we measured the acquisition cost of RP4 in varying growth conditions. To do so, we chose alternative conditions such that conjugation occurred in an environment where excess glucose resulted in decreased growth efficiency. Then, we supplemented the conjugation media with increasing concentrations of casamino acid (CAA) to restore growth efficiency. We observed a clear dose response relationship between casamino acid levels and acquisition costs (Figure 3D); maximum costs were observed when cell growth was most inefficient, strengthening our argument that metabolic adaptation modulates this phenomenon. Moreover, to rule out any confounding effect of varying growth rates, which increase with CAA alongside metabolic efficiency, we also included a low-growth/high efficiency condition; results indicated that changes in acquisition costs were independent of growth rate. We have updated the text to discuss these experiments:

“Given that RP4 and pR-specific differences likely arise due to differences in energetic demand, we hypothesized that altering growth efficiency (e.g., the amount of substrate consumed that is converted to biomass) (Chudoba *et al.*, 1992), would modulate acquisition costs. Intuitively, inefficiently growing cells generate excess available energy (Russell, 2007; Russell & Cook, 1995) that may be readily applied to plasmid-related metabolic demands, potentially resulting in a lower acquisition cost. In contrast, efficiently growing cells devote the bulk of available energy to biomass production (Low & Chase, 1999), and thus, reallocating that energy to plasmid demands may increase acquisition costs. To test this hypothesis, we focused on modulating growth conditions. It is well-established that excess glucose yields highly inefficient *E. coli* growth (Basan *et al.*, 2015; Liu, 1998), but that efficiency is restored with exogenous amino acid supplementation (Waschina *et al.*, 2016; Akashi & Gojobori, 2002). Adapting a recent approach that leveraged this trade off (Lopatkin *et al.*, 2019), we quantified plasmid acquisition costs under three casamino acid (CAA) concentrations (0, 0.01, and 0.1% w/v) and excess glucose (0.4% w/v). Since higher CAA increases both efficiency and growth rate, we included a fourth combination (0.04%/0.1% w/v glucose/CAA) where growth rate is comparable to 0.4%/0.01% w/v, but results in higher efficiency due to the lower glucose level (Fig. 3C). In accordance with our intuition, increased CAA resulted in significant decreases in acquisition costs (Fig. 3D; $P < 0.05$, one-tailed t-test, see Table S3B for exact P-values). Moreover, this trend was not an effect of increased growth: acquisition cost in 0.04%/0.1% glucose/CAA was significantly less than in 0.4%/0.01% glucose/CAA conditions ($P=0.04$, one-tailed t-test). Together, these results suggest that environmental conditions significantly modulate acquisition cost through changes in growth efficiency.”

2. Increased expression upon first entry in the cell, and possible reasons for this could be elaborated on. Specifically, one phenomenon not discussed here is transitory derepression of plasmid conjugative transfer genes, which leads to high rates of transfer after first entering the cell. Transitory derepression and its consequences for plasmid and host fitness have been modelled previously (Lundquist and Levin, Genetics 1986; Dionisio, Evol Ecol Research 2005), as a strategy for plasmid

epidemic spread while minimizing costs. Importantly, it makes plasmids highly transferable for the first generations after transfer, making the assumption that conjugation from SD is negligible in the model inappropriate. This will likely affect the consequences on plasmid population dynamics.

Some mechanistic data could help provide some insight into what is responsible for this acquisition cost. Transcriptomics comparing RP4 recent and established transconjugants could pinpoint the plasmid or host genes involved; plasmids with derepressed conjugation are also available, and could maybe be used here: for instance, F plasmid or R1 derepressed mutants (see Haft et al, ISME 2009) might have a high fitness cost that does not disappear, whereas repressed versions could lead to a high acquisition cost but lower fitness one.

These are incredibly interesting and insightful questions. Indeed, in many cases, transient derepression of plasmid-encoded machinery occurs immediately following plasmid acquisition. To address this possibility, we explored the impact of transitory derepression in-depth using both experimental comparisons of plasmids with and without derepression, as well as our computational model. We pursued the suggestion to investigate R1 and other *incF* plasmids, including both their fitness and acquisition costs. In line with the Reviewer's intuition, we found that the mutant derepressed R1drd plasmid does impose a *fitness* cost on the host, but that its native repressed counterpart, R1, does not (Appendix Fig. S5). Likewise, the other derepressed *incF* plasmid, pRK100, also imposed a small fitness cost on the host (Appendix Fig. S5). However, despite this variability in fitness costs, none of the three *incF* plasmids tested were found to impose an acquisition cost. Thus, despite the Reviewer's hypothesis that transitory depression may influence acquisition cost, R1 and R1drd behaved quantitatively and qualitatively similarly. To further bolster this point, we investigated two additional plasmids with and without a repression system: R6K (naturally repressed) and R6Kdrd (derepressed mutant) belong to the *incX* group. Both of these plasmids exerted an acquisition cost; as with R1/R1drd, there was no significant difference between this repressed/derepressed pair. Moreover, because none of the *incF* plasmids we tested imposed an acquisition cost, we suggest that *inc-* to *inc-*group differences may influence acquisition costs. Our final findings and hypothesis are written in the text as follows:

“We next looked at acquisition costs across a broad panel of plasmids to further investigate its generality. Specifically, we first quantified acquisition costs for six well-characterized conjugal plasmids (R1, *incF*; R1drd, *incF*; R6K, *incX*; R6Kdrd, *incX*; pRK100, *incF*; RIP113, *incN*; Appendix Table S1B) covering four additional incompatibility groups and a range of fitness costs (Appendix Fig S5). Importantly, two pairs of these plasmids represent derepressed mutants and their native repressed counterparts (R1 and R6K); although both plasmid types express conjugation machinery immediately following their transfer, repressed plasmids tightly regulate machinery expression shortly thereafter, minimizing its fitness cost (Lundquist & Levin, 1986) (Appendix Fig S5). Interestingly, we observed no qualitative differences in R1 or R6K acquisition costs, regardless of conjugation repression (Fig. 3E): both R6K/R6Kdrd, and neither R1/R1drd, were costly to acquire. Since all variants express machinery immediately upon entry, these results suggest that machinery gene repression occurs on a timescale longer than that of acquisition cost for these four plasmids. Rather, we noted acquisition cost differences across incompatibility groups. Specifically, all four *incF* plasmids (R1, R1drd, and pRK100), along with pR from earlier, imposed no significant acquisition cost, whereas R6K and R6Kdrd (*incX*) and RIP113 (*incN*) induced a strong acquisition cost. Incompatibility groups are differentiated by their plasmid replication/partitioning mechanisms as well as specific copy number (Kittell & Helinski, 1993); for example, *incF* plasmids typically exist at low copy numbers ($\leq \sim 2$) (Burger *et al*, 1981), whereas *incX* plasmids can be present at 10-15 copies per cell (Rakowski & Filutowicz, 2013). Thus, these results suggest that acquisition costs may arise as a consequence of establishing plasmid-specific replication and maintenance mechanisms.”

Given the lack of difference observed between derepressed and repressed plasmids, and the extensive testing on various plasmid types, we felt that further robust transcriptomics-based investigations were not critical at this point, though they represent interesting possibilities for future work. Instead, these results led us to investigate more about the role of the cellular metabolism, manifesting in the experiments described in point 1.

3. Overall, I think some restructuring of the data could make the main points of the paper more convincing, even without additional data. Some of the interesting data are actually a bit buried under many controls, or not exploited as much as they could, appearing more anecdotal than they deserve. An extreme example is the effect of recipient strain. Having data on *Klebsiella* is great and indeed generalises the results, but then the BW25113 strain is really not needed (at least not in the main figure) as it is almost identical to MG1655. In Figure 3, the effects of strain BW25113 and dilution on acquisition cost could be moved to supplementary results, or at least combined in a single graph. On the other hand, figure 3F is very interesting and should be given more space and discussion!

We thank the reviewer for these helpful suggestions. Given our additional included data, the previous figure layout required even more restructuring than the Reviewer had originally suggested. In particular, we have now added results with seven additional plasmids [clinical vs well-characterized] and various additional growth conditions. Specifically, Figure 3 has been expanded to include a range of generality experiments, including growth rate and acquisition cost under variable CAA/glucose as previously described (Figure 3C-D), acquisition cost of six well-characterized plasmids (Figure 3E), and the acquisition cost of an additional *incI* clinical plasmid not investigated in the original submission. Moreover, the *Klebsiella* results were moved to Figure 2 (2E) to enhance the discussion of RP4's acquisition cost; the discussion of those results was also expanded as suggested:

“Moreover, RP4 was costly to acquire for the clinically-isolated recipient strain *Klebsiella pneumoniae* (KPN), indicating species-level generality (Fig 2E). The drastic difference in RP4 acquisition costs between *E. coli* and KPN recipients suggest that cost is not solely a function of particular plasmids; strain/species level attributes are likely key as well.”

Finally, the data on the effects of strain BW25113 was moved to the supplementary results as suggested.

4. The existing data could also be exploited more to obtain more quantitative results. For instance, could the authors estimate the number of generations during which the acquisition cost is present from their results?

This is yet another highly insightful question. Ideally, the generation time associated with the acquisition cost should be quantified at the single cell level. However, as all of our data was obtained in bulk populations, we unfortunately do not feel that our results lend themselves to quantifying the generation time. The Fernandez-Lopez *et. al* 2014 paper previously mentioned by the Reviewer, which noted depressed transconjugant growth rates shortly following conjugation and a significantly longer first generation time, did measure the total duration of these effects in terms of the number of generations. However, their study is significantly different from ours in that they emphasize the role of transcriptional overshoot post-conjugation and the long-term effects of plasmid fitness costs. Specifically, they hypothesize that a transient increase in gene expression would translate to a higher energetic burden, and thus, longer generation time. In contrast, our study focuses on the phenotypic consequences of plasmid acquisition; we quantify the acquisition cost in terms of transiently slower growth rates and longer lag times, which we show are compensated for within 24 hours. Moreover, our quantification methods (CFU counts and plate reader growth) preclude us from longitudinally tracking individual cells over time; this would be necessary to quantifying the duration of acquisition effects, either in units of time or generations. However, we fully agree with the Reviewer that such quantification would be of interest in future works.

5. I have one main methodological issue: could some of the delay in growth be due specifically to the requirement for expression of plasmid-carried resistance genes with antibiotic selection? In that case this would not strictly represent a cost of plasmid acquisition but a delay in plasmid benefits, and could be seen as an artefact of the method used here.

We agree with the Reviewer that the expression of resistance genes is one potential explanation of the observations under our experimental setup. However, we note that our expanded data involving additional plasmids provides additional confidence that antibiotic resistance gene expression plays a negligible role in the observed phenotype. In particular, none of the four *incF* plasmids studied (pR, pRK100, R1, R1drd) imposed an acquisition cost, despite the fact that all express antibiotic resistance and that several of these experiments were conducted under identical antibiotic conditions that led to acquisition costs with other plasmids. For example, as shown in Appendix Table S1B, conjugation of pRK100 was conducted under Carb (100 ug/mL) / Spec (50 ug/mL) conditions, and resulted in no acquisition cost. On the other hand, both R6K and R6Kdrd conjugation experiments were performed in identical antibiotic conditions and *did* result in an acquisition cost. Furthermore, in both cases, these are large, self-transferrable plasmids. Thus, though we cannot completely rule out the possibility that resistance delays contribute to acquisition cost, we are confident there are additional, more significant factors at play. We summarize these findings in the text:

“Given the dual antibiotic selection in our experimental setup, we note that observed acquisition costs may arise due to a delay in corresponding resistance gene expression. However, we identified several cases in which no acquisition costs were observed, despite transconjugants being subject to identical dual antibiotic selection, e.g., low metabolic efficiency conditions and for various *incF* plasmids. Moreover, acquisition costs were variable within specific dual antibiotic combinations. These results suggest that, although resistance expression may contribute to acquisition costs, it is not the primary determinant.”

Minor points:

1. There is a problem with reference numbering, with several references duplicated.

We appreciate the reviewer bringing this to our attention and have updated the references accordingly.

2. page 2 definitions: strictly speaking, cell-to-cell contact would include other means of HGT like nanotubes, not only conjugation. Similarly, the cell fusion shown in Figure 1A drawing is confusing when representing conjugation.

We have updated the fused donor/recipient cell schematic in Fig. 1A to more accurately depict the conjugation process.

3. Page 5: 'For each panel ...' should go in the figure legend

We have moved the aforementioned sentence into the legends for Appendix Fig. S2 and Appendix Table S2.

4. Page 7 'mobile plasmids' are mobilisable plasmids?

The term “mobile plasmid” has been updated to “mobilizable plasmids” in all cases.

5. page 9: ref 27 is not about a model of conjugation.

Reference 27 was used to indicate the method employed for growth rate quantification, namely the Baranyi method; this is an established equation used to estimate both growth rate and lag time, and indeed is not specific to conjugation. To ensure that our method of growth quantification did not result in any artifacts, we verified that other established methods (including the standard logistic equation, the Gompertz equation, and a custom-written algorithm), resulted in the same statistical conclusions (Appendix Fig. S2).

6. page 14: from 'more generally', the end of results might be more suited to the discussion

We have updated the manuscript to move the aforementioned section on page 14 into the Discussion section. It can now be found at the end of the first paragraph:

“Our results update our canonical interpretation of growth dynamics during conjugation: they may not be fully defined by a static fitness cost. Indeed, when acquisition costs are particularly large, disregarding their effect can result in inaccurate persistence predictions (Appendix Fig S8), which is often a key objective in modeling *in vivo* microbial communities and/or microbial risk assessment.”

7. there are several minor mistakes in the writing, of which I have probably missed some: for instance e.g. is used for referring to explanations not examples; page 22 'the prensky method ...' sentence lacks a verb.

We updated the manuscript to address these issues. Cases where “e.g.,” was used for explanations have been switched to “i.e.,” and all other grammar has been carefully reviewed and updated as appropriate. We updated the specific sentence in question to:

“The Prenskey method was designed using curve-smoothing and numerical differentiation to identify the time, t , at which the maximum derivative occurred within the region of exponential growth.”

8. page 23: I am no expert in this but more details should probably be given on the model fitting details, than 'using an optimisation function'- I would not know how to replicate this from these methods.

We thank the reviewer for bringing this to our attention and have made changes to the methods section that we believe strengthen the paper and make the techniques used clearer to scientists who want to replicate our experiments. We have included further detail about how to reproduce the optimization function in MATLAB; the paragraph now states:

“In all cases for data fitting, β was calculated using the `fminsearch` function in MATLAB. `fminsearch` is an optimization function that minimizes any user-specified objective function, and requires an initial estimate of the parameter(s) to be fitted. In our case, we implemented this fitting by minimizing the difference between the ODE solution of $S^D + S^A$, and the raw data curves as shown in Appendix Fig. S7; since ρ was constrained by experimentally estimating the ratio of growth rates between adapted and *de novo* populations, β was the only remaining free parameter to be fitted. Thus, for each plasmid, the initial β estimate was set to be the geometric lag time of the corresponding growth curve, and the output of the `fminsearch` optimization was an estimated β that best fit our experimental data.”

9. Across the manuscript, full p-values should be stated (not only < 0.05).

We have updated the manuscript to state the full p-values where appropriate. We also include a supplementary table of p-values that we reference throughout the manuscript. These values can be found on Appendix Table S3.

10. page 6 (and all related figures): why not use the time to threshold itself as a measure of acquisition cost, instead of predicted cell densities? It would seem more intuitive to me, a higher cost directly leading to a longer time to grow instead on calculating the discrepancy between predicted and actual cell densities for initial transconjugants (which are not the factor actually modified here)

We agree with the Reviewer that the time to threshold is a more intuitive representation of the acquisition cost when taken in isolation. However, given the emphasis on multiple plasmids and diverse growth environments, the initial number of cells does indeed vary. Moreover, presenting acquisition costs in terms of initial cell numbers allows the reader to quickly develop an intuition as to the underlying conjugation process. For example, it is clear from our presentation that both R1 and R1drd had the highest conjugation efficiency

due to the greatest true CFU, whereas pRK100 has a much lower conjugation efficiency. This additional information would be lost if we presented only differences in time-to-threshold. Moreover, the relationship between time-to-threshold and initial cell density varied significantly across plasmids, conditions, and strains used; thus, actual cell densities represent a consistent basis upon which to compare and interpret results across experiments. As a result, we did not feel the time-to-threshold on its own was the optimal way of representing all of the data combined. Thus, we opted to keep the presentation of acquisition costs in terms of predicted versus actual cell densities, and hope the Reviewer understands our rationale. We have updated the text to include the following sentence:

"Moreover, the use of T_0 as a metric of acquisition cost, rather than t^* , allows us to simultaneously estimate conjugation rates across diverse experimental conditions and plasmids."

11. do the authors have a hypothesis about why there is high variation in individual replicates in Figure 1C? Are the recently acquired transconjugants always more variable than the adapted ones?

This is an excellent observation and one that we have discussed at great length! It does indeed seem that initial transconjugants exhibit higher variability than adapted ones. Our best explanation of this trend is that the fewer number of cells starting in the well leads to more apparent day-to-day and batch-to-batch variability. However, this variability very well may be due to plasmid-specific factors, such as the number of pili that are initially expressed or the effects of establishing copy number control. Moreover, we note that conjugation efficiencies varied by plasmid, as represented in Figs 3B, 3E, 3F, 3G, etc. by the number of transconjugants produced. In some cases, one hour of conjugation yielded 10^8 transconjugants, whereas in other cases, only 10^4 were produced.

12. Figure S6B could be plotted in log scale as well, as the initial dynamics is interesting and cannot be seen here.

We have included an additional panel in Fig. S6B using a log-scaled y-axis.

Reviewer #2:

This is a clear and interesting paper reporting "acquisition costs" of plasmids, that are neatly distinguished from fitness costs. The paper combines well-designed and controlled experiments with mathematical modelling to develop a full description of acquisition costs for a focal plasmid-host association and then expands this to other plasmid-host associations to prove the generality of the phenomenon. Finally, they show that a new augmented model including acquisition costs and fitness costs better explains longer term plasmid dynamics especially where acquisition costs are appreciable.

This is a well-written and presented study. The paper is very clear and easy for follow and the work builds logically upon the previous work by the group. The discovery of acquisition costs is novel and interesting, and the inclusion of multiple plasmid-host associations reveals some very intriguing variability in the magnitude of acquisition costs.

Given the high quality of the work and its presentation I do not have very many critical comments. I think the discussion would benefit from a few words speculating why acquisition costs might vary among plasmid-host associations. Another potentially interesting subject for discussion may be whether acquisition costs would be under selection e.g. through plasmid evolution to minimise these. Also, a key missing reference that would add to the discussion of compensatory evolution dynamics is [Hall et al 2020 Microbiology] which shows the potential for very rapid compensatory evolution.

We are grateful for the Reviewer's positive feedback and are encouraged that they too find these results as exciting as we do. We agree the suggested discussion topics improve the overall context of the paper, and have updated the text accordingly in the discussion section:

“We also demonstrate that the acquisition cost is general to four incompatibility groups. Tellingly, all *incF* plasmids studied here were uniquely found not to impose an acquisition cost. Given that incompatibility groups employ different replication/partitioning mechanisms, often in conjunction with specific host gene expression, it can be reasonably assumed that the acquisition cost may vary among plasmid-host combinations. However, we show that acquisition costs arise in different recipient strains *and* different recipient species, further proving the generality and scope of this phenomenon.”

We have also included the Hall *et al.* reference, as suggested, in our background on compensatory mutations.

14th Dec 2020

RE: MSB-20-9913R, Conjugation dynamics depend on both the plasmid acquisition cost and the fitness cost

Dear Dr. Lopatkin,

Thank you again for sending us your revised manuscript. We have now heard back from reviewer #1 who was asked to evaluate your study. As you will see below, this reviewer is satisfied with the modifications made and is supportive of publication. They recommend some minor (and optional) text modifications, which could be included in a revised version.

Moreover, before we can formally accept the manuscript for publication we would ask you to address a few remaining editorial issues listed below:

- Our data editors have noticed some unclear or missing information in the figure legends, please see the attached .doc file. Please make all requested text changes using the attached file and *keeping the "track changes" mode* so that we can easily access the edits made.
- Please include a callout to Figure panel 1F in the main text.
- The Reagents and Tools Table is not formatted correctly. More information as well as downloadable templates (.doc or .xls) for the Reagents and Tools Table can be found in our author guidelines: . An example of a Method paper with Structured Methods can be found here: .
- I noticed that you have provided Figure Source Data as a single .xls file. In this format it is not clear to which figures or figure panels this data refers to. Source Data for main figures should be provided in .zip Folders labeled "Source data for Figure X". Please provide one .zip folder for each of the main figures. Source Data for Appendix Figures should all be provided in one single .zip folder labeled "Source Data for Appendix". Further information regarding Source Data can be found here: . In case it is too complicated to split the file into separate ones (e.g. because the same data is used in multiple figures/panels) you can leave it in a single .xls, labeled and called out in the text as Dataset EV1. Please include a separate tab with a short description of the dataset. Dataset EV1 should be called out in the Data Availability section.
- The synopsis and bullet points text was rather long and I have tried to streamline it. Could you please let me know if you agree with the edited version attached below?

Please resubmit your revised manuscript online, with a covering letter listing amendments and responses to each point raised by the referees. Please resubmit the paper ****within one month**** and ideally as soon as possible. If we do not receive the revised manuscript within this time period, the file might be closed and any subsequent resubmission would be treated as a new manuscript. Please use the Manuscript Number (above) in all correspondence.

Click on the link below to submit your revised paper.

Link Not Available

Yours sincerely,

Maria Polychronidou, PhD
Senior Editor
Molecular Systems Biology

If you do choose to resubmit, please click on the link below to submit the revision online before 13th Jan 2021.

IMPORTANT:

8. Please note that corresponding authors are required to supply an ORCID ID for their name upon submission of a revised manuscript (EMBO Press signed a joint statement to encourage ORCID adoption) (<https://www.embopress.org/page/journal/17444292/authorguide#editorialprocess>).

Currently, our records indicate that the ORCID for your account is 0000-0003-0018-9205.

Please click the link below to modify this ORCID:
Link Not Available

The system will prompt you to fill in your funding and payment information. This will allow Wiley to send you a quote for the article processing charge (APC) in case of acceptance. This quote takes into account any reduction or fee waivers that you may be eligible for. Authors do not need to pay any fees before their manuscript is accepted and transferred to the publisher.

*** PLEASE NOTE *** As part of the EMBO Press transparent editorial process initiative (see our Editorial at <https://dx.doi.org/10.1038/msb.2010.72> , Molecular Systems Biology will publish online a Review Process File to accompany accepted manuscripts. When preparing your letter of response, please be aware that in the event of acceptance, your cover letter/point-by-point document will be included as part of this File, which will be available to the scientific community. More information about this initiative is available in our Instructions to Authors. If you have any questions about this initiative, please contact the editorial office (msb@embo.org).

Reviewer #1:

The authors have addressed all main points of my previous review, and I believe the ones of the

other reviewer too. They performed additional experiments, both with several other plasmids, and in additional experimental conditions, which extends the generality of their results and allows them to back up their hypotheses about causes of plasmid acquisition cost. I really liked this new version and believe it is suitable for publication in *Molecular Systems Biology*.

I have only one last comment, but this is really only semantics and the authors can disagree! I see acquisition costs as a component of fitness costs, not an alternative effect, and would suggest reformulating some parts of the text, for instance 'independently of fitness effects' to 'independently of other fitness effects', etc. In my mind, plasmid fitness will depend on two main components: plasmid horizontal transmission and plasmid effects on host fitness, and both acquisition and longer-term fitness costs would ultimately be parts of the latter, whereas opposing acquisition costs to fitness costs seems to suggest a third, conceptually different effect.

Reviewer 1

The authors have addressed all main points of my previous review, and I believe the ones of the other reviewer too. They performed additional experiments, both with several other plasmids, and in additional experimental conditions, which extends the generality of their results and allows them to back up their hypotheses about causes of plasmid acquisition cost. I really liked this new version and believe it is suitable for publication in *Molecular Systems Biology*.

We are pleased to read that the Reviewer is satisfied with our revisions and is supportive of the publication of our manuscript. We are grateful for the Reviewer's time and feedback.

I have only one last comment, but this is really only semantics and the authors can disagree!

I see acquisition costs as a component of fitness costs, not an alternative effect, and would suggest reformulating some parts of the text, for instance 'independently of fitness effects' to 'independently of other fitness effects', etc. In my mind, plasmid fitness will depend on two main components: plasmid horizontal transmission and plasmid effects on host fitness, and both acquisition and longer-term fitness costs would ultimately be parts of the latter, whereas opposing acquisition costs to fitness costs seems to suggest a third, conceptually different effect.

We thank the reviewer for raising this point. Indeed, the difference between fitness and acquisition effects was a major question as we generated additional results. While in some cases it does seem as though the acquisition cost is potentially related to the fitness cost (such as with RP4), there are also cases where the acquisition cost appeared primarily unrelated (e.g., R1drd and pR). Moreover, plasmid p193 exhibited a *beneficial* fitness cost but a disadvantageous acquisition cost, consistent at two different conjugation temperatures. Due to the many different combinations of fitness costs and acquisition costs, and in taking the reviewer's view into consideration, we have updated relevant statements to the following: "This plasmid acquisition cost occurs independently of long-term fitness effects". We believe this both accurately captures our results, and acknowledges the types of fitness effects noted by the reviewer.

13th Jan 2021

RE: MSB-20-9913RR, Conjugation dynamics depend on both the plasmid acquisition cost and the fitness cost

Dear Allison,

Thank you again for sending us your revised manuscript. We are now satisfied with the modifications made and I am pleased to inform you that your paper has been accepted for publication.

*** PLEASE NOTE *** As part of the EMBO Publications transparent editorial process initiative (see our Editorial at <https://dx.doi.org/10.103/msb.2010.72>), Molecular Systems Biology publishes online a Review Process File with each accepted manuscripts. This file will be published in conjunction with your paper and will include the anonymous referee reports, your point- by-point response and all pertinent correspondence relating to the manuscript. If you do NOT want this File to be published, please inform the editorial office at msb@embo.org within 14 days upon receipt of the present letter.

LICENSE AND PAYMENT:

All articles published in Molecular Systems Biology are fully open access: immediately and freely available to read, download and share.

Molecular Systems Biology charges an article processing charge (APC) to cover the publication costs. You, as the corresponding author for this manuscript, should have already received a quote with the article processing fee separately.

Please let us know in case this quote has not been received.

Once your article is at Wiley for editorial production, you will receive an email from Wiley's Author Services system, which will ask you to log in and will present you with the publication license form for completion. Within the same system the publication fee can be paid by credit card, an invoice or pro forma can be requested.

Payment of the publication charge and the signed Open Access Agreement form must be received before the article can be published online.

Upon acceptance it is mandatory for you to return the completed payment form. Failure to send back the form may result in a delay in the publication of your paper.

Molecular Systems Biology articles are published under the Creative Commons licence CC BY, which facilitates the sharing of scientific information by reducing legal barriers, while mandating attribution of the source in accordance to standard scholarly practice.

Proofs will be forwarded to you within the next 2-3 weeks.

Thank you very much for submitting your work to Molecular Systems Biology.

Best wishes and Happy New Year,

Maria

Maria Polychronidou, PhD
Senior Editor
Molecular Systems Biology

Corresponding Author Name: Dr. Allison J. Lopatkin

Manuscript Number: MSB-20-9913